# Illuminating the complete ß-cell mass of the human pancreas- signifying a new view on the islets of Langerhans

Joakim Lehrstrand [1], Wayne I. L. Davies [1], Max Hahn[1], Olle Korsgren[2], Tomas Alanentalo[1] & Ulf Ahlgren [1] ✉

Pancreatic islets of Langerhans play a pivotal role in regulating blood glucose homeostasis, but critical information regarding their mass, distribution and composition is lacking within a whole organ context. Here, we apply a 3D imaging pipeline to generate a complete account of the insulin-producing islets throughout the human pancreas at a microscopic resolution and within a maintained spatial 3D context. These data show that human islets are far more heterogenous than previously accounted for with regards to their size distribution and cellular make up. By deep tissue 3D imaging, this in-depth study demonstrates that 50% of the human insulin-expressing islets are virtually devoid of glucagon-producing α-cells, an observation with significant implications for both experimental and clinical research.

By virtue of their roles in regulating blood glucose homeostasis, the hormone producing islets of Langerhans have been the subject of intense research for well over a century[1]. Still, understanding their distribution and intra-islet organization across the human pancreas remains limited. In humans, estimates of islet numbers, volumes and cellular ratios vary significantly. Inter-individual biodiversity apart, this may be attributed to differences in selected analytical methods, most of which rely on extrapolation of a limited amount of two-dimensional (2D) data, or on subfractions of isolated islets. The non-diabetic (ND) human pancreas has been reported to contain between 1–14.8 million islets[2–4] and with reports of mean diameters of ~109 μm[4], 130 μm[3,5] or even above 300 μm in older literature[6], whereas the total islet volume has been estimated to between 0.5–2.0 cm$^3$ (see ref. 2 and references therein). Similarly, analyses of the relative contributions of different endocrine cell-types to overall islet composition display significant differences. A consensus of several reports suggest that human islets are composed of ~60% insulin (INS)-producing ß-cells and 30% glucagon (GCG)-producing α-cells (ref. 4 and refs therein). The remaining 10% primarily consists of somatostatin-producing δ-cells, followed by pancreatic poly-peptide (PP) and ghrelin-producing ε-cells, respectively (ref. 2 and references therein). With regards to overall distribution, the islet mass is suggested to be both unevenly[7] and evenly[4] distributed within the pancreas.

Islet cells require intercellular communication to function properly, including a large number of paracrine signals that act within islets together with exogenous neural inputs to ensure correct islet responses to glucose and other metabolites[8,9]. It is generally acknowledged that diabetes is a disease that involves all islet cell-types, not only ß-cells[10]. Therefore, a precise understanding of islet organization throughout the entire organ (including size, distribution, and hormonal composition) is critical to fully appreciate the significance of islet architecture and whole organ pancreatic distribution for normal physiology and disease etiology. In this study, mesoscopic optical 3D imaging approaches[11] were used to provide a whole organ account of the β-cell mass distribution (i.e., islet volume and number, as well as 3D spatial location) across the entire volume of the human pancreas. It has previously been demonstrated that the human islet population is heterogenous with regards to cellular architecture and composition[12,13], and different subtypes of endocrine cells even exist within individual islets[13,14]. Data presented here, derived from five ND donors (see Supplementary Table S1) provides convincing evidence for a previously unknown heterogeneity in islet composition, in that as much as 50%

[1]Department of Medical and Translational Biology, Umeå University, Umeå, Sweden. [2]Department of Immunology, Genetics and Pathology, Uppsala University, Uppsala, Sweden. ✉e-mail: Ulf.Ahlgren@umu.se

of human islets contain only a few (<1%) or no glucagon-producing α-cells. Apart from the direct implications for pre-clinical and clinical research, these observations will serve as a foundation for the generation of precise anatomical atlases of the human pancreatic endocrine system and how it is affected under pathological conditions.

## Results

### 3D reconstruction of the complete ß-cell distribution of the human pancreas

Recently, the labeling and imaging of cm³-volumes of human pancreatic tissue at a microscopic resolution was demonstrated, while maintaining the 3D context of all labeled cells[15]. Implementing adapted protocols (see methods), the entire β-cell distribution throughout the pancreas of a deceased ND donor was analyzed (see Supplementary Table S1). Briefly, donated pancreata were divided into tissue disks, using a 3D printed matrix (Fig. 1A and Supplementary Fig. S1) that were labeled for insulin and scanned individually by near infrared optical projection tomography (NIR-OPT)[16,17] (Fig. 1B–F, Supplementary Fig. S2, Supplementary Movie S1). Selected regions of interest (ROIs) were also scanned by higher resolution light sheet fluorescence microscopy (LSFM)[18,19]. For NIR-OPT data, an INS⁺ object in this study is defined as a distinct body of cells stained for insulin that cannot be separated from each other at the resolution observed during segmentation (here, OPT scan settings results in an isotropic resolution of ~21 μm). By aligning the resultant tomographic datasets together in 3D space, an entire pancreas could be reconstructed with regards to the 3D distribution of INS⁺ cells (Fig. 1G, Movie S2 and Supplementary Fig. S3). Segmentation of INS⁺ signals allowed for the extraction of a complete picture of all INS⁺ objects including their individual volumes and spatial 3D coordinates throughout the volume of the pancreas. Hereby, a range of comprehensive statistical assessments of the β-cell mass (BCM), or more precisely β-cell volume, distribution could be extracted within a maintained spatial 3D context that is not dependent on the extrapolation of partially sampled data.

### Revealing the normal distribution of the human β-cell mass - an absolute volumetric assessment

The ND donor pancreas displayed in Fig. 1 (H2457, see Supplementary Table S1) comprises an INS⁺ volume of $1.17 \times 10^{12}$ μm³ (=1.17 cm³) that consists of $2.21 \times 10^6$ separate INS⁺ islets at the applied resolution. To regionally refine these values, the pancreas was divided into four regions consisting of the head (region 1, using the indentation of the superior mesenteric vein as a boundary) and regions 2–4 consisting of portions 1/3 in length of the remainder of the pancreas. These analyzes showed that the ß-cell density is relatively uniform across the length (head to tail) of the human pancreas (Fig. 2A, B), although variations exist between individual disks, possibly due to regional differences in vascular and ductal densities (Fig. 2A). Analyses of one disc each from region 1–4 from four additional pancreas (H2456, 2466, 2506 and 2522) are displayed in Supplementary Fig. S4.

As islet size has important functional implications (see discussion below), the $2.21 \times 10^6$ measured INS⁺ objects were subdivided into different size categories, where the fraction that each size category represented as a volume normalized to the entire tissue volume was calculated (Fig. 2D). To better understand islet size distribution, further calculations were performed that included the volume fraction out of total INS⁺ volume (Fig. 2E), INS⁺ objects per size category per mm³ (Fig. 2F) and the fraction of INS⁺ objects constituted by each size category (Fig. 2G). Similar data were obtained from all regions (1–4), alluding to a relatively uniform size distribution of INS⁺ islets across the length of the human pancreas (Supplementary Fig. S5). To simplify comparisons of these 3D volumetric measurements to previous morphometric 2D studies, each volume size category was converted to the diameter of a spherical object for a particular corresponding range interval (Fig. 2C). Note, however, that islets are not perfectly spherical objects (See Supplementary Fig. S6 for sphericity analyzes of iso-surfaced volumes). Presented data show that around 75% of the total BCM consists of INS⁺ islets in the range of $400 \times 10^3$–$12,800 \times 10^3$ μm³ (which corresponds to spheres ~91–290 μm in diameter), but that this volume only corresponds to ~26 % of all INS⁺ islets (see blue and red

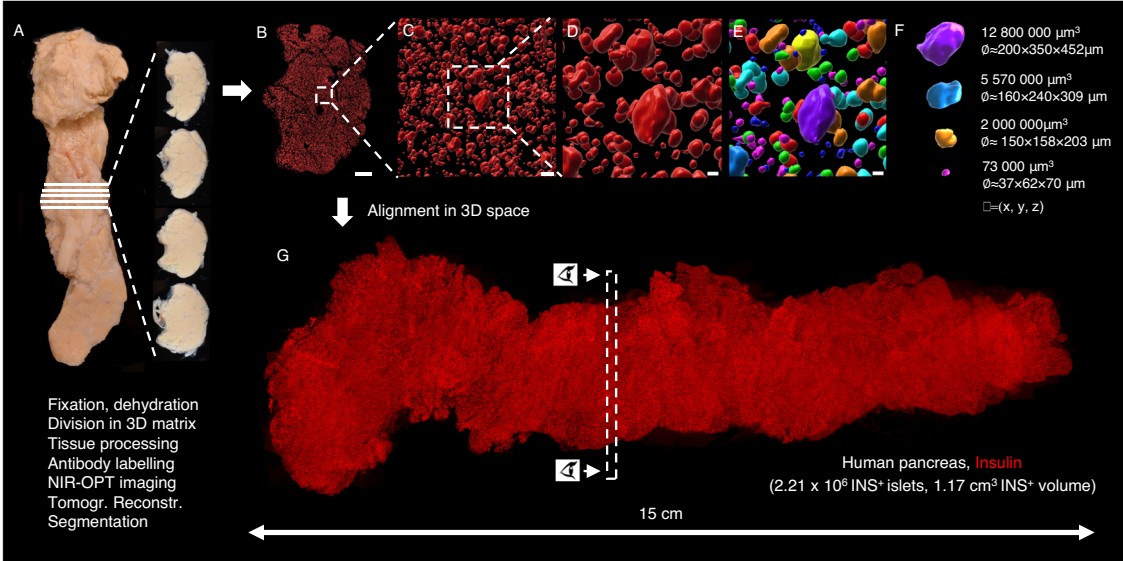

12 800 000 μm³
Ø≈200×350×452 μm

5 570 000 μm³
Ø≈160×240×309 μm

2 000 000μm³
Ø≈ 150×158×203 μm

73 000 μm³
Ø≈37×62×70 μm

☐=(x, y, z)

Alignment in 3D space

Fixation, dehydration
Division in 3D matrix
Tissue processing
Antibody labelling
NIR-OPT imaging
Tomogr. Reconstr.
Segmentation

Human pancreas, Insulin
(2.21 x 10⁶ INS⁺ islets, 1.17 cm³ INS⁺ volume)

15 cm

**Fig. 1 | Generation of volumetric and 3D-spatial data sets of the complete β-cell distribution across the human pancreas.** The intact pancreas (**A**) was fixed directly after procurement from a deceased donor (H2457, see Supplementary Table S1) and dehydrated in 100% ethanol. Using a 3D printed matrix, the pancreas was divided into 2.8 mm disks, with known spatial origins, allowing for penetration of tissue processing chemicals and antibodies. Each disc was scanned individually by NIR-OPT (see Supplementary Movie S1) or LSFM before offline segmentation and 3D rendering of individual islet β-cell volumes (**B–F**). In (**E**), each pseudo-colored object corresponds to a specific β-cell volume with known 3D coordinates. **G** By combining individual tomographic datasets in 3D space, new data sets could be created that encompass every insulin-stained object: in this case there were $2.21 \times 10^6$ INS⁺ objects with a total volume of 1.17 cm³ ($n = 1$ donor pancreas subdivided in $n = 51$ pancreatic tissue disks), here displayed as a single 3D maximum intensity projection (see also Supplementary Movie S2 and Supplementary Fig. S3). Eye symbols in (**G**) illustrates the angle of view of disc shown in (**B**). Scale bar in (**B**) is 3000 μm, in (**C**) 200 μm and in (**D, E**) 70 μm.

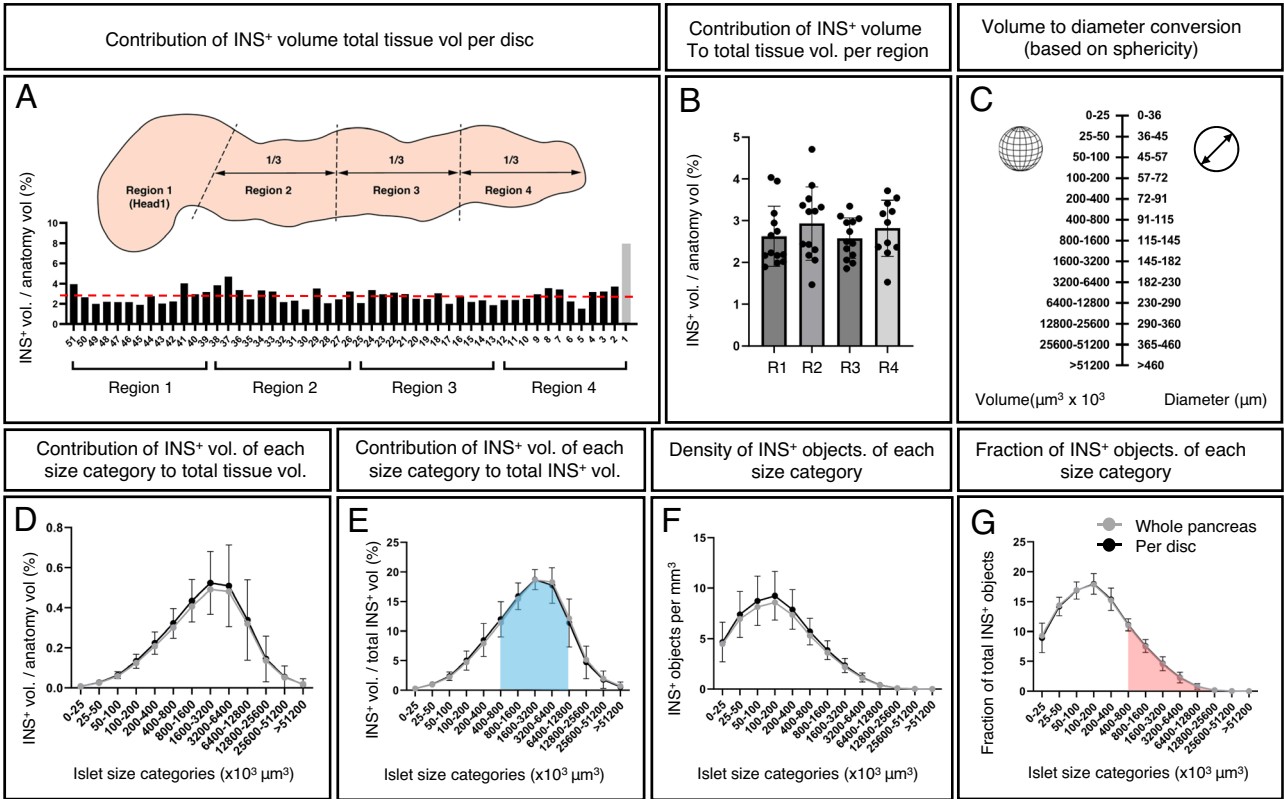

**Fig. 2 | 3D quantification and statistical assessment of the complete human pancreatic β-cell distribution. A** INS+ volume normalized to the tissue volume for each disc (H2457, see Supplementary Table S1). Note, whereas the walls of vessels and ducts are included in the total tissue volume, the lumens of these structures (i.e., empty space) are not. The average INS+ density across the entire pancreas is 2.8% (red broken line). Schematic inset illustrates the regional separation of the pancreas into four defined regions. **B** A pooled bar graph showing the same data as in (**A**) redefined for each of the four regions of the human pancreas. **C** A graph illustrating the corresponding diameter of spherical objects for each size category (for comparison to 2D stereological studies), noting that islet sphericity is not perfect, and this value is not the same as the average 3D diameter (see Supplementary Fig. S6). Graphs showing the volume of each INS+ islet size category represented as (**D**) volumetric density normalized to the entire tissue volume, (**E**) volumetric density normalized to the entire INS+ volume, (**F**) number of INS+

objects per mm³ per size category and (**G**) the fraction of INS+ objects per INS+ islet size category (**G**). Black lines in (**D**–**G**) represents average values per disc and gray lines represents values based on all INS+ islets normalized against the entire pancreatic tissue volume. The blue area in (**E**) corresponds to the red area in (**G**), showing that the majority of the BCM (~75%) consists of only ~25% of islets. Gray bar (disc 1, R4) indicates a very small tissue disc at the very tip of the tail that encompassed only 573 INS+ objects (see Supplementary Fig. S1). GraphPad robust regression and outlier removal (ROUT) with Q = 1% identified this small sample as an outlier and it was removed from subsequent analysis. $n = 1$ donor pancreas subdivided in $n = 51$ pancreatic tissue disks. Particularly large disks were subdivided to accommodate scanning, $n = 75$ in (**D**–**G**). Note, the resolution limit of the scanner at the implemented zoom is 21 μm. Error bars show mean ± SDE. Source data are provided as a Source Data file.

color-coding in Fig. 2E, G, respectively). Further, when all pancreatic islets are considered, these data indicate that the average islet diameter (based on insulin staining in 3D space) decreases to ~Ø65 μm (or ~Ø68.3 μm when accounting for typical tissue shrinkage (here ~5% between fixed and cleared tissue, see Supplementary Fig. S7) and perfect sphericity (i.e., Ψ = 1)). In this report, all values are given without compensation for this factor. Of note, it was previously demonstrated that at the current resolution, using OPT, the average 3D-diameter of islets, based on the insulin signal, deviated less than ±5% compared to the islet diameter measured on Hematoxylin/Eosin stained tissue sections of the same islets[15].

**Human and mouse islets have a similar contribution of islet size categories to the overall BCM but differ in their organization within the organ**

As demonstrated by optical 3D imaging, rodent islets are heterogeneously distributed both within and between the three primary, splenic, duodenal and gastric, pancreatic lobes[20–22], and analyses of different diabetes disease models showed that islets of different sizes can be unequally affected during disease progression[23–26]. E.g., in the Non Obese Diabetic (NOD) model for naturally induced T1D, smaller islets are more susceptible to autoimmune destruction, whereas large

centrally located islets even appear to possess a compensatory growth potential[23]. However, similar types of assessments of the human pancreas have been effectively hindered by technological limitations, and our knowledge about the normal distribution of the human β-cell mass in 3D space is limited.

By defining small medium and large sized islets as making up 1/3 each of the total BCM normalized to the entire tissue volume, calculated from one disc from each region (1–4) in five donor pancreata (H2456, H2457, H2466, H2506 and H2522, see Supplementary Table S1) and cross reference this information back to the 3D reconstruction of each disc we could not detect any pronounced heterogeneities in the distribution of islets of different size-categories within the organ (Fig. 3A–D, Supplementary Fig. S8 and Supplementary Movie S3). Possibly, larger islets could appear less frequent in the absolute periphery of the organ, but this observation was not consistent across all samples/regions (Supplementary Fig. S8). Whereas the relative contribution of each size category to the overall BCM is similar between mice and humans (compare Fig. 3D, H), the spatial organization of islets of different size categories differ. In analogy with previous reports[20,21], large mouse islets were preferentially located in vicinity of the organ's central axis, following the main pancreatic duct, whereas no such pattern was clearly discernable in the human disks

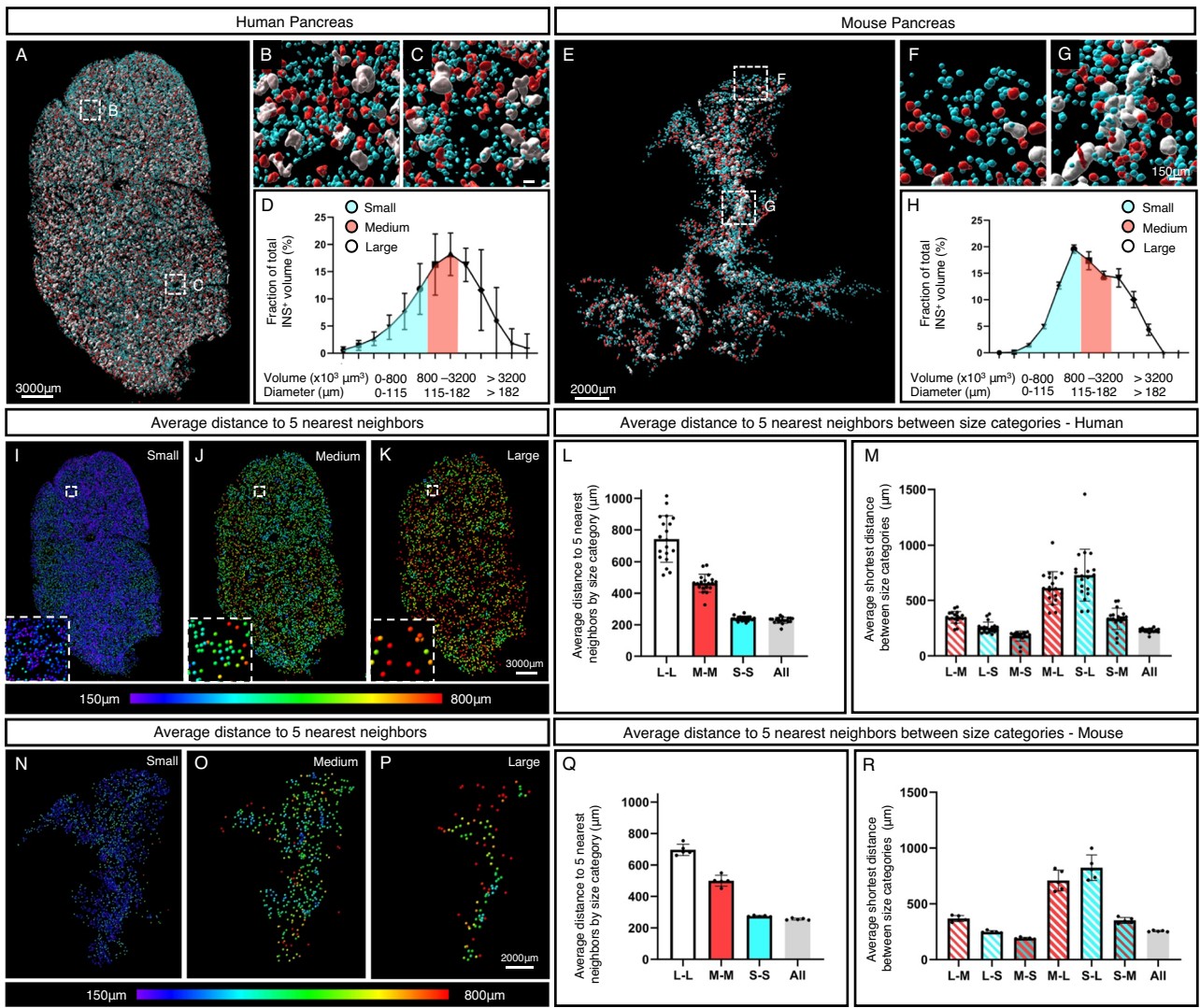

**Fig. 3 | Spatial arrangement and relative contribution of islet size categories in man and mice. A–C** Images of a representative pancreatic disc in which the insulin signal has been segmented and pseudo colored according to size. Each size category (blue, small (S); red, medium (M); white, large (L)) corresponds to 1/3 of the total β-cell volume. **D** Graph showing the volume of each INS⁺ islet size category represented as volumetric density normalized to the entire tissue volume, calculated from one disc from each region (1–4) in five donor pancreata (H2456, H2457, H2466, H2506 and H2522, see Supplementary Table S1). **E–G** Images of a representative mouse pancreas pseudo colored as in (**A**). **H** Graph showing the volume of each INS⁺ islet size category represented as volumetric density normalized to the entire tissue volume, calculated based on all INS⁺ islets in C57BL/6 pancreata at 10 weeks (*n* = 5, including 28108 INS⁺ islets). **I–K** Heat maps showing the average distance to the 5 nearest objects for each size category in the disc depicted in (**A**). Bar graphs showing the average distance to the 5 nearest neighboring objects (islets) by size category (**L**) and between size categories (**M**) calculated from one disc from each region (1–4, *n* = 20 tissue disks) from five donor pancreata (*n* = 5, H2456, H2457, H2466, H2506 and H2522, see Supplementary Table S1). "All" denotes the distance to the 5 nearest neighbors regardless of size category. GraphPad robust regression and outlier removal (ROUT) with Q = 1% identified H2422, R4, as an outlier and it was removed from the analyses. **N–P** Heat maps showing the average distance to the 5 nearest objects for each size category in the mouse pancreas depicted in (**C**). Bar graphs showing the average distance to the 5 nearest neighboring objects (islets) by size category (**Q**) and between size categories (**R**) based on all INS⁺ islets in C57BL/6 pancreata (*n* = 5). Note, the resolution limit of the scanner at the implemented zoom is 21 μm. Error bars show mean ± SDE. Source data are provided as a Source Data file.

(compare Fig. 3A, E and Supplementary Movie S3 and S4). Heat maps, visualizing the average distance of each islet to its 5 nearest neighbors within each size category, confirmed this picture (compare Fig. 3I–K, N–P) and in contrast to previous stereological assessments we could not find convincing evidence for islet routes[4] or varying islets densities in specific areas of the human pancreas, head to tail[7]. For both species, the average distance to the 5 nearest neighbors was longest for the large islets and shortest for the smallest, reflecting the relative number of islets within these size categories (Fig. 3L, M, Q, R).

Altogether, this global assessment of human pancreatic β-cells at the current level of resolution provides strong evidence that the BCM is relatively evenly distributed along the pancreas from head to tail, that the average islet size based on insulin staining, is significantly smaller (Ø65 μm or Ø68.3 μm corrected for shrinkage) than what has been reported by 2D stereological studies[2–6] or by assessments of isolated islets (ref. 27 and references therein), and that the human β-cell mass organization differ significantly from that of the mouse. Further, they open for precise 3D dimensional, combined volumetric, assessments of specifically antibody targeted cells and structures throughout the volume of human donor pancreata in different pathological settings.

## The majority of the human INS⁺ islets are devoid of GCG expressing cells

It is generally accepted that a direct interaction between α-cells and β-cells is crucial for glycemia management[28]. As a first step in exploring

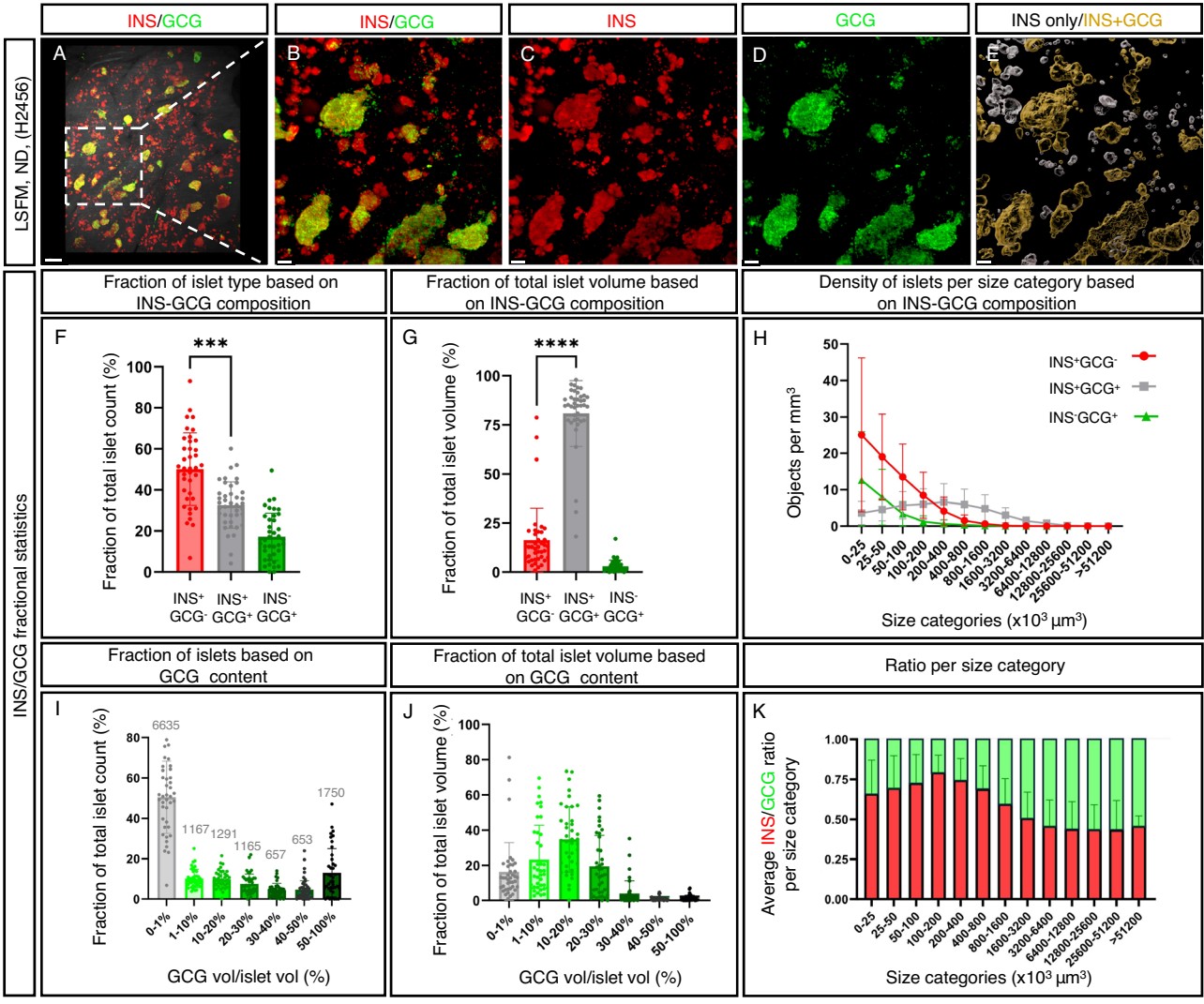

**Fig. 4 | High resolution deep tissue 3D imaging reveals prominent heterogeneities in human islet INS- and GCG-cell composition. A–D** 3D Maximum intensity projection (MIP) from a LSFM scan of a representative ROI (see methods) from a ND donor pancreas stained for insulin (INS, red) and glucagon (GCG, green) showing abundant presence of INS⁺GCG⁻ islets. **E** Pseudo-colored transparent surface reconstructions where islets with <1% GCG⁺ volume are white and islets with >1% GCG⁺ cells are orange (see also Supplementary Movie S5). **F** A graph showing the average fraction of INS⁺GCG⁻, INS⁻GCG⁺ and INS⁺GCG⁺ islets. **G** A graph showing the contribution of islets with <1% GCG⁺ volume to the total volume of INS- and GCG-expressing cells. **H** Densities of INS⁺GCG⁻ (red line), INS⁺GCG⁺ (gray line) and INS⁻GCG⁺ (green line) in (**F**) and (**G**) represented as a function of size category. **I** GCG volume/islet volume as fraction of total islet count per ROI. **J** GCG volume/

islet volume as fraction out of total islet volume. **K** Average ratio of INS/GCG per size category. For individual specimen in (**F**) and (**G**) see Supplementary Fig. S12 and for (**K**), see Supplementary Fig. S13. Data are derived from regions 1–4 (two ROIs/region, *n* = 40 ROIs) from ND donor pancreata (*n* = 5, Supplementary Table S1 and Supplementary Fig. S11), scanned at 1.9 µm x-y spatial resolution. Note, a filter was introduced in the volume quantification to exclude INS⁺ islets corresponding to spherical objects with a diameter of less than 29 µm (see methods). Scale bar in (**A**) is 500 µm and scale bars in (**B–E**) are 100 µm. **F** two-tailed paired *t*-test, ***(*p* = 0.0002) and **G** two-tailed Wilcoxon matched-pairs signed rank test ****(*p* < 0.0001). Error bars show mean ± SDE. Source data are provided as a Source Data file.

this relationship in the entire human pancreas, 3D OPT-based imaging approaches for large scale studies of intra-islet cellular composition were applied to representative tissue disks from regions 1–4 from ND pancreata encompassing both genders and an age range of 20–45 years of age (*n* = 5, see Supplementary Table S1). 3D maximum intensity projection (MIP) views (Supplementary Figs. S9 and S10), covering the entire depth of tens of thousands of islets for each scan, revealed a dramatic degree of heterogeneity in intra-islet composition that is not consistent with the consensus 2:1 ß-cell to α-cell ratio (refs. 2,4 and references therein). Instead, a very high proportion of human islets, primarily in the size range below 800 × 10³ µm³ in volume (≈90 µm in diameter), appeared to be completely devoid of GCG altogether or possess just a few α-cells (Supplementary Figs. S9 and 10D, E). To confirm this observation, and to resolve the possibility that individual

GCG⁺ cells were not included at the applied resolution when using OPT (~21 µm), LSFM scans of ROIs within the same tissue disks at a spatial resolution of 1.9 µm were generated. Hereby, cellular ratios across the volume of the investigated tissue disks at a high cellular resolution could be determined (Fig. 4A–E, Supplementary Movie S5 and Supplementary Figs. S11–13). Analyzes of two ROIs from each of the four anatomical regions in five different ND pancreata (see Supplementary Table S1) revealed that on an average 50.2% of the INS⁺ islets contain <1% GCG-expressing (GCG⁺) cells. These islets, however, although abundant in numbers, corresponded to only ~16% of the total islet volume (Fig. 4F–H). Serial 2D sections stained for INS and GCG, including every section of individually assessed islets identified by LSFM imaging, supported these observations of the presence of a significant number of INS⁺GCG⁻ islets in the human pancreas

(Supplementary Fig. S14). Specifically, these data indicate that most GCG$^+$ cells reside within larger islets (>800 × 10$^3$ μm$^3$ in volume; >115 μm in diameter) and that the absolute majority of the GCG$^+$ volume is comprised by islets containing 1–30% GCG$^+$ cells, with 10–20% GCG$^+$ cells making the greatest contribution (Fig. 4I–K). Altogether, these studies provide evidence for a dramatic heterogeneity in islet size and composition in the human pancreas, with a majority of, chiefly smaller, islets consisting of INS$^+$, but not GCG$^+$ cells.

## Discussion

Already in 1906, Lydia Dewitt stated "No organ or tissue of the body has been the subject of more thought or investigation than have the areas of Langerhans, especially during the past years"[29]. Given the ongoing diabetes pandemic and the immense efforts still active in both basic and clinical diabetes arenas, this statement still holds true over a century later. This present report provides an in depth whole organ account of the complete islet ß-cell mass distribution within the human pancreas, while maintaining the spatial 3D context (i.e., volume, shape and location) of every single islet. Assuming that every islet contains β-cells expressing insulin (INS$^+$), this study has detected and analyzed 2.21 × 10$^6$ INS$^+$ islets in a representative healthy ND donor pancreas constituting around 2.8% of the pancreatic volume. This value is in the upper range of previous reports of 1–3% (See e.g., refs. 30–33). Unlike earlier studies, these measurements are not dependent on extrapolation of a limited amount of 2D or 3D data. Although BCM density may vary between individual disks, our analysis suggests that BCM is relatively uniformly distributed across the analyzed organ, head to tail, when the entire tissue volume is analyzed (see Fig. 2A). Although inter-individual differences may be noted in BCM density and islet size distribution, analysis of complementary material (one disc each from region 1–4 from four additional donors) show no clear trend towards higher BCM in the tail region (Supplementary Fig. S4). Given that the current study points to local variations between disks, each covering the 3D volume of tens of thousands of islets, we suggest that undertakings aimed at measuring BCM density based on limited amounts of 2D data may become quite sensitive for even very local variations in islet distribution. It is of course possible that other parameters such as age, BMI, ethnicity etc., together with differences in analytical methods and the amount of material analyzed and how it has been normalized may account for differences between different reports on islet distribution and cellularity.

This study further suggests that the average human islet is significantly smaller than previously reported[2–6], which has important sampling and functional implications as discussed below. It should be noted hereby, that our OPT derived value (65 μm or 68.3 μm corrected for tissue shrinkage) is based on the average 3D diameter of the islet's actual 3D shape, whereas previous in situ data are predominantly based on cross sectional diameters. These values are in turn dependent on both islet shape and the cross-sectional level of the analyzed islet. In all, our measurements of islet sizes describe a seemingly gaussian distribution, like previous reports (see e.g., ref. 3). However, our data is somewhat skewed towards smaller islets, probably due to differences in analysis techniques in that more "small" islets are detected. It has previously been suggested that small islets contain a relatively large number of INS$^+$ cells (63%) compared to large islets where the number of INS$^+$ cells is lower (39%), and that these smaller islets have greater insulin secretion[34]. The deep tissue imaging approach applied in this report, which covers the entire tissue volume, show that very few islets display the literature consensus composition of around 60% INS$^+$ and 30% GCG$^+$ cells[2]. Instead, the human endocrine pancreas comprises ~50% INS$^+$ islets, which are essentially devoid of GCG$^+$ cells. These are however predominantly found in the smaller end of the spectrum of islet size ranges (still larger than 29 μm in diameter). Noteworthy, the investigated pancreata contained between ~50.2 and 63% INS$^+$GCG$^-$ islets, with the exception of H2506 which contained ~30% INS$^+$GCG$^-$

islets (Supplementary Fig. S12 and Supplementary Table S1), still a considerable figure. The large number of INS$^+$GCG$^-$ islets observed here raises important questions about islet specialization and functionality. A large number of paracrine signals have been demonstrated to act within islets to ensure their proper functional responses to glucose and other metabolites[35]. Further, studies imply that heterologous contact between α-cells and ß-cells are crucial for glycemia management[28]. Most significantly, insulin (INS) together with other signaling molecules inhibit glucagon (GCG) secretion, whereas glucagon stimulates insulin secretion. Cholinergic innervation to human islets is sparse[36]. However, α-cells have been reported to provide ß-cells with acetylcholine, which enhances insulin release by binding to muscarinic receptors[36], and it has been shown that blocking GCG and GLP-1 signaling in mice limits insulin secretion[37]. As such, it is very surprising that the vast number of INS$^+$ islets devoid of GCG (i.e., INS$^+$GCG$^-$ islets) reported here has not been recognized previously. This may be due, until now, to the lack of an accurate and sensitive 3D imaging perspective, which allows for the simultaneous study of multiple islets throughout a large tissue depth. It is also possible that a general assumption that islets have a typical mixed cell population has eluded their discovery and subsequent reporting in most 2D stereological studies. For example, when assessing cryosection or wax embedded samples with islets lacking GCG$^+$-cells, it has been plausible to assume that GCG$^+$-cells would be present on a consecutive slide or that the observed islet was not representative. Whereas isolated islets have become an invaluable tool to understand islet biology and diabetes[38], data presented here emphasize that human islet isolates do not reflect the full range of islets sizes and cellular compositions in the endogenous islet population. In addition, islets distributed for research are commonly associated with cultivation artifacts[39]. This is of particular significance since in vitro studies, of rat and human islets, suggest that smaller islets have superior cellular function compared with larger islets[34,40,41]. Interestingly, it has also been demonstrated that small mouse islets after isolation are virtually devoid of α-cells[42], and perhaps this subset of murine islets represent the subpopulation of human INS$^+$GCG$^-$ islets identified in this present study.

Whereas insulin is key for glucose metabolism, glucagon primarily controls amino acid metabolism[43]. The observed heterogeneity in islet composition suggests that pancreatic islets of different cellular arrangements might promote specialized functional niches in order to meet different metabolic demands. For example, it is possible that there is an endogenous plasticity in islet constitution in response to prolonged exposure to specific metabolites. As such, it is plausible that ß-cells within islets with different proportions of hormonal cells may have different metabolic activities. Further studies incorporating other hormonal cell-types and markers for different cellular subtypes may shed light on these questions, as well as to the possibility that islets of different configurations and putative functional roles are distinctly innervated.

Altogether, the results presented here advocate for a re-evaluation of how an islet of Langerhans is defined. Further, they strongly suggest that islet size and cellular composition should be carefully considered in any research or clinical setting aiming at reflecting or reproducing the endogenous islet population of the human pancreas. As such, these data are likely to greatly impact a broad range of undertakings in islet and pancreas research, including personalized design and bioengineering approaches for ex vivo experimentation of isolated islets to strategies for islet generation of islets by stem-cell or bio-printing approaches, as well as the optimization of clinical islet transplantation protocols that better reflect the endogenous microcellular environment of the pancreas. The imaging platforms utilized in this study are under continuous development, including e.g., increased field of view, shorter scan times, better computational processing protocols etc. Although OPT and LSFM provide complementary information, it should be understood that the

data outputs that are not directly comparable. E.g., whereas OPT generates isotropic voxels, the implemented LSFM scanner displays elongation effects in the Z-axis due to the inherent principal of detection. As for example, smaller volumes of labeled cells, otherwise under a defined threshold, could be included in the islet count. Nevertheless, the developed possibility to study the human pancreas using specific antibody labeling, from all angles and through its entire depth (delivering 3D coordinates, volumes and shapes for all labeled objects) will provide a unique opportunity to identify features of normal pancreatic anatomy and disease pathologies that would otherwise be extremely challenging to recognize using other techniques. Finally, the developed approach applied in this study was able to overcome the frequent issue of reagent penetration in optical imaging of larger tissues. As such, the methods presented here should be translatable to a plethora of studies that target a variety of other human and non-human tissues, potentially in any physiological and/or pathophysiological context, with essentially a full freedom of target selectivity.

## Methods

### Ethics declaration

All work involving human tissue was conducted in accordance with the Declaration of Helsinki[44] and in the European Council's Convention on Human Rights and Biomedicine[45]. Consent for organ donation for use in research was obtained from the donor prior to death via the Swedish National Donor Registry (https://www.socialstyrelsen.se/en/apply-and-register/join-the-swedish-national-donor-register/) or from relatives of the deceased donors conferred by the donor's physician and documented in their medical records. The study was approved by the Regional Ethics Committee in Uppsala, Sweden (Dnr 2017/1471-32, 2023-01845-01).

### Organ isolation and processing

Pancreata were obtained after declared brain death from 5 multi-organ donors registered within the framework for the Nordic Network for Clinical Islet Transplantation (NNCIT), with clinical data relevant to this study listed in Supplementary Table S1. The donated pancreata were dissected from the duodenum in Ringer's acetate solution (Braun Melsungen, Germany) and washed in 1 × PBS (Medicago, Sweden) before fixation in 4% formaldehyde (Solveco, Sweden). After 24 h, the fixative was replaced with fresh 4% formaldehyde solution and fixed for another 24 h. Fixed samples were then stepwise dehydrated in ethanol: twice with 75% (v/v) ethanol in $H_2O$ and twice with 96% (v/v) (VWR, Sweden) at 4 °C. At this stage, pancreata were stored and transported in 96% (v/v) ethanol at room temperature (RT). In order to remove residual fat, pancreata were extensively washed with 96% (v/v) ethanol at 4 °C on an orbital shaker, replacing with fresh ethanol solution daily until the alcohol became transparent (i.e., no excess fat droplets present in the samples). Subsequently, all organs were imaged using a Nikon D5200 camera (see Supplementary Fig. S1).

### Tissue preparation for 3D imaging

A slicing matrix was designed with a section thickness of 2.8 mm, providing an optimal balance between reagent penetration and resolution/magnification to distinguish individual insulin-positive (INS⁺) objects for thresholding and quantification (Tinkercad, Autodesk, USA), using a 3D printer (Prusa i3 MK3S, Prusa Research, Czech Republic). Pancreata were mounted in 1.5% (w/v) low melting temperature agarose (Cat. No. 50100; Lonza, USA) at 37 °C in the custom-made 3D-printed matrix (Supplementary Fig. S1). Subsequently, whole pancreata were cut into 2.8 mm thick "slabs" and the agarose carefully removed. Tissue slabs wider than 2 cm (in the head region) were cut in two to ensure a good fit in the field of view when applying near infra-red optical projection tomography (NIR-OPT) microscopy (see below). Images were taken during the slicing process to document X, Y and Z

coordinates for each slab (Supplementary Fig. S1). Tissue slabs were then stored in 100% methanol (MeOH; Cat. No. 67-89-4; Fisher Scientific, Sweden) at −20 °C until further processing.

### Whole-mount immunohistochemistry staining and tissue clearing

Tissue slabs were initially taken from −20 °C to −80 °C in 100% MeOH for at least 1 h, then cycled between −80 °C to RT at least 5 times (typically for 2–2.5 h per step) to increase antibody penetration. To bleach pigmented cells and reduce autofluorescence, each sample was incubated for 12 h (or overnight) in bleaching solution containing hydrogen peroxide solution ($H_2O_2$; Cat. No. H1009; Sigma-Aldrich, Merck, Germany) at a final concentration of 15% (v/v), dimethyl sulfoxide (DMSO; Cat. No. D5879; Sigma-Aldrich, Merck, Germany) and MeOH ($H_2O_2$:DMSO:MeOH) in a ratio of 3:1:2. After a further 8–12 h incubation in fresh bleaching solution, samples were washed twice in 100% MeOH at 30–60 min per wash at RT.

For whole-mount immunolabeling, all samples were rehydrated stepwise (33%, 66% and 100%, 1 h per step) from MeOH/TBST (0.15 M NaCl, 0.1 M Tris-HCl and 0.1% Triton® X-100 (Cat. No. 108603; Sigma-Aldrich, Merck, Germany), pH 7.5), followed by a further wash with 100% TBST for 1 h. Next, slabs were incubated at 37 °C in blocking solution comprising TBST supplemented with 10% heat-inactivated goat serum (Cat. No. CL1200-500; Cedarlane, Canada), 5% DMSO (Cat. No. D5879; Sigma-Aldrich, Merck, Germany) and 0.01% sodium azide ($NaN_3$) for 2 days. After blocking, samples were incubated for 7 days at 37 °C with primary antibodies diluted in blocking solution after first being filtered through a 25 mm Acrodisc® 0.45 µm syringe filter (Cat. No. 4614; Pall Corporation, USA) to remove any potential artifacts such as aggregated antibody complexes. Primary antibodies used consisted of guinea pig anti-insulin/pro-insulin (INS; Cat. No. 16049; Progen Biotechnik, Germany; diluted 1:3000) and rabbit anti-glucagon/pro-glucagon (GCG; Cat. No. HPA036761; Atlas Antibodies, Stockholm, Sweden; diluted 1:1000). Following incubation with primary antibodies, all samples were rigorously washed three times in TBST heated to 37 °C on a rotator for 1 h per wash step, then overnight, followed by two washes the next day in TBST as before. Next, samples were incubated at 37 °C for 5 days with filtered secondary antibodies in blocking solution, including donkey anti-guinea pig IRDye® 680 (Cat. No. 926-68077; Li-Cor Biosciences, USA; diluted 1:250) and donkey anti-rabbit Alexa Fluor® 594 (Cat. No. 711-585-152; Jackson ImmunoResearch, UK; diluted 1:500). Once again, antibody solutions were filtered to remove any potential fluorophore precipitates. After secondary labeling, all slabs were washed three times in TBST on a rotator (at 37 °C; 1 h per wash step, then overnight), followed by two washes the next day in TBST.

For clearing and subsequent imaging, slabs were mounted in 1.5% low melting point agarose at ~37 °C, cooled to set at RT for ~30 min, then moved to 4 °C to completely solidify for at least 3–4 h, preferably overnight. All mounted tissue slabs were cut to remove excess agarose to form clean, straight edges without any acute angles and subsequently dehydrated in aqueous MeOH solutions, moving stepwise from 33%, to 66% and finally 100% (1 h per step). Mounted samples were then dehydrated through at least 5 cycles of 100% MeOH (to remove all traces of $H_2O$) with each step conducted overnight with gently shaking. When fully dehydrated, slabs were optically cleared using a 1:2 mixture of benzyl alcohol (Cat. No. 109626; VWR, Sweden) and benzyl benzoate (BABB) (Cat. No. 105860010; Acros Organics, USA), which was replaced every 12–24 h at least 3–5 times until optimally cleared before imaging.

### Near infra-red optical projection tomography (NIR-OPT) and light sheet fluorescence microscopy (LSFM) imaging

Mounted human pancreatic slabs were scanned submerged in BABB as previously described[15,46] in a custom-built near-infrared optical

projection tomography (NIR-OPT) scanner[16], constructed using a Leica MZFLIII stereomicroscope (Leica, Germany) with a CoolLED pE-4000 LED fluorescence light source (Ludesco Microscoped, USA) and an Andor iKon-M (Andor Technology, UK) camera with a tilted mirror and a step motor for sample rotation. A zoom factor of 1.25× was used for all samples, rendering an isotropic voxel size of ~21 μm. Filter sets (with stated excitation (Ex) and emission (Em) parameters) used were as follows: insulin (INS), Ex: HQ 665/45 nm, Em: HQ 725/50 nm; glucagon (GCG), Ex: 565/30 nm, Em: 620/60 nm; and anatomy (auto-fluorescence, AF), Ex: 425/60, Em: 480 nm LP.

To verify the presence of glucagon negative (GCG⁻) islets, selected pancreatic slabs stained for INS and GCG, and previously scanned by NIR-OPT, were reimaged at higher resolution using an UltraMicroscope II (Miltenyi Biotec, Germany), which comprised a 1× Olympus objective (Olympus PLAPO 2XC) coupled to an Olympus MVX10 zoom body, a 3000 step chromatic correction motor and a lens corrected dipping cap MVPLAPO 2× DC DBE objective. Samples were scanned at 1.6× magnification with a numerical aperture of 0.141, a depth of 1000 μm with a 5 μm stepsize, giving a voxel size of $1.9 \times 1.9 \times 5$ μm, with a dynamic focus set to 10 images across the field of view. Light sheets were merged using the built-in projection function. Filter sets used were as follows: INS, Ex: 650/45, Em: 750/60; GCG, Ex: 580/25, Em: 625/30; and anatomy (AF), Ex: 470/40, Em: 525/50, where an exposure time for all channels was kept at 300 ms. The resultant datasets were saved in *ome.tif format native to ImSpectorPro software (version 7.1.15; LaVision Biotec, Germany), before being converted into 3D projection *ims files using Imaris File Converter software (version 10.0.0; Bitplane, UK).

### Post-NIR-OPT image pre-processing, 3D reconstruction and surfacing

For post-NIR-OPT processing and 3D reconstruction, all generated images were processed using the same pipeline as follows: (i) in order to increase the signal to noise (S:N) ratio within each image, the pixel range of acquired NIR-OPT frontal projections was adjusted individually to display minima and maxima; (ii) a contrast limited adaptive histogram equalization (CLAHE) algorithm was applied to equalize large contrast differences in immunofluorescence[21], with a tile size of 48 by 48 with sharpening for all signal channels; (iii) the axis of rotation was centered computationally post-scanning using a Discrete Fourier Transform Alignment algorithm (DFTA) from the open-access DSP-OPT software package[47,48] (https://github.com/ARDISDataset/DSPOPT); (iv) processed datasets were reconstructed to tomographic sections using NRecon software (version 1.7.0.4, SkyScan Bruker microCT, Belgium) with added misalignment compensation and ring artifact reduction; (v) all resultant *.bmp images were converted to *.ims file format as before using Imaris File Converter software.

Using Imaris software (version 10.0.0; Bitplane, UK), 3D surfaces of reconstructed OPT scans for anatomy (AF), INS⁺ and GCG⁺ signal were generated using a built-in automated batch processing pipeline with manually adjusted threshold levels where needed, applying both a gaussian blur and background subtraction (largest object: 158 μm) both to reduce noise, including objects with a voxel value above 5. To correct for the signal gradient in insulin staining, a two-step surfacing method was applied. Firstly, a surface to effectively segment peripheral high intensity objects was generated. Secondly, an additional surface aiming to segment low intensity objects was added and included a filter to exclude overlapping objects (with a minimum 10 μm³ overlap) compared to the original segmentation. The two resultant 3D volumes were then merged to account for objects including an intensity gradient (Supplementary Fig. S1). Finally, artifacts outside the tissue volume (determined by AF) were manually excluded.

Quality control of the surfacing method was determined by manually counting islets containing β-cells in 3 regions of interest (ROIs) for 3 slabs (H2457), in ImageJ (version 1.53k, National Institutes of Health, USA) for signal versus final segmentation, taking into consideration 7069 islets in the analyzes (see Supplementary Fig. S2 and Supplementary Table S2). Accuracy was evaluated as a percentage of segmentation count vs signal count. This surfacing approach showed high levels of conformity to manual counting; therefore, the pipeline was applied to the analyzes of the whole pancreas.

### 3D alignment of the whole human pancreas

To fully reconstruct the whole human pancreas (see Fig. 1 and Supplementary Fig. S3), individual tissue slab 3D *ims datasets were imported and manually aligned in Imaris 3D space. Specifically, each slab was sequentially oriented based on the original anatomical images, where autofluorescent large ducts and vessels were used to guide accurate alignment, until the whole pancreas consisting of 51 tissue slabs was completely reconstructed.

### Data size and time consumption

NIR-OPT projection views of an entire pancreas constituted approximately 96 GB of data per channel (Insulin and anatomy respectively), in total 192 GB of *.tif images, and was scanned in approximately 170 h. The post processed files, with intermediate steps, constituted approximately an additional 768 GB of data. Tomographic reconstruction of the resulting files generated approximately 145 GB of *.bmp files. These were reconstructed in approximately 1 week. Volumetric rendering and analysis files constituted in total 93 GB of *.ims files. Finally, manual 3D alignment of individual *.ims files (for each disc) was estimated to roughly 3–4 working days.

### Assessment of insulin islet 3D distribution and organization

To determine islet spatial organization in 3D space, the INS⁺ surfaces were first divided into three size categories (Fig. 3) in Imaris, corresponding to equal fractions of the total INS⁺ volume. Surfaces were then converted into spots by first applying the distance transformation function which effectively generates a channel based on the surface volumes. The resulting channels were then the basis for generating spots that more accurately correspond to the initial surface segmentation. The spots function was used to evaluate average distance to neighboring objects within categories, as well as determining the average shortest distance between size categories. In the spot function, distances are calculated between the center of mass of any two objects. A similar pipeline for comparison between human and mouse INS⁺ islet distribution was carried out on five 10-week-old male C57BL/6J control mice from a pre-processed and segmented public dataset[24]. Isolation and processing of these mouse pancreata is described in refs. 24,47 (and references therein). Division into size categories was performed as for the human dataset.

### Determining insulin and glucagon composition in OPT and LSFM

To evaluate the frequency of GCG⁻ islets in OPT scans of human pancreatic slabs (Supplementary Figs. S9 and S10), 3D surfaces were generated using Imaris software of the total number of INS⁺ and GCG⁺ objects in 4 slabs (one taken from each of the four pancreatic regions) from 5 donors as described above. In addition, to generate surfaces representing all islets (independent of INS and GCG composition), INS⁺ and GCG⁺ surfaces were duplicated as GCG⁺ objects were seemingly larger. Therefore, surfacing GCG⁺ more accurately reflect the islet border. In this case, all INS⁺ surfaces were subjected to an overlap exclusion filter of a minimum 10 μm³ overlap with GCG⁺ volumes. The two resulting surfaces were then merged to represent all islets. Sub-surfaces were generated from this collection of total islet 3D volumes depending on the ratio overlap of INS⁺ and GCG⁺ surfaces; these were defined into three classes as follows: (i) INS⁺GCG⁺, (ii) INS⁺GCG⁻, and (iii) INS⁻GCG⁺, where negative surface signal was

defined as being less than 1% of immunolabelling for any given endocrine cell-type to account for background noise.

To determine islet INS and GCG ratios in LSFM scans (Fig. 4 and Supplementary Figs. S11–13), two types of surfaces were similarly generated using Imaris software, one aimed at distinguishing the islet border and the other to segment INS⁺ and GCG⁺ cells, respectively (Supplementary Fig. S16). These analyzes were performed on ROIs sized $1.7 \times 1.7 \times 1$ mm. As part of the pre-processing pipeline, all channels were subjected to layer normalization in Imaris software. Surfaces of stained INS⁺ and GCG⁺ objects were generated with an added background subtraction of 10 μm and exclusion of all objects below $3 \times 10^3$ μm³ to remove noise. Moreover, to effectively segment the border of pancreatic islets, INS and GCG channels were surfaced using absolute intensity with a smoothing texture detail 3.78 μm, where the two resultant surfaces were merged to better define the entire islet (Supplementary Fig. S16). Note, since signal intensity may vary depending on where each ROI was located on the horizontal axes of each specimen (due to the configuration of the light sheets in the scanner), thresholds were carefully set manually (assessing both over- and under exposed values) to reflect the borders of the labeled cells. Further, an exclusion filter was applied for objects $<1.4 \times 10^4$ μm³ to only include objects with a minimum diameter of 29 μm (assuming sphericity) from the analyzes therefore excluding single cells from the analysis.

### Post 3D imaging histology and validation of glucagon negative islets

Slabs derived from different regions of the pancreas stained for insulin and glucagon, then 3D scanned with OPT followed by LSFM, were washed several times in 100% MeOH to remove all traces of BABB, followed by stepwise rehydration in a series of 70%, 50%, 30% and 10% (v/v) ethanol to $1 \times$ PBS for 1 h at RT with gentle shaking per step. Agarose was removed by first washing the slabs in 0.29 M sucrose (Cat. No. 10319003; Fisher Scientific, Sweden) in $1 \times$ PBS at 57 °C, then two further washes with 0.29 M sucrose solution at RT with careful manual removal of agarose where required. After agarose removal, tissues were cryoprotected by incubating in 30% (w/v) sucrose in $1 \times$ PBS overnight at 4 °C to prevent the formation of ice crystals during the freezing process, embedded and snap frozen in NEG-50 (Cat. No. 11912365; Fisher Scientific, Sweden), and stored at −80 °C. 10-um-thick sections were collected onto SuperFrost Plus glass slides (Cat. No. 10149870; Fisher Scientific, Sweden), air dried at RT and washed in TBST for 10 min. Tissue sections were blocked in 10% fetal bovine serum (FBS; Cat. No. 11550356; Sigma-Aldrich, Merck, Germany) for 1 h at RT and re-stained with insulin (diluted 1:10000) and glucagon (diluted 1:5000) in blocking solution at RT overnight. Slides were washed $3 \times 5$ min in TBST and incubated with 4′,6-diamidino-2-phenylindole (DAPI) and the following secondary antibodies in blocking solution for 2 h at RT: Alexa Fluor 488® goat anti-guinea pig IgG H&L (Cat. No. ab150185; Abcam, UK; diluted 1:500 and Alexa Fluor 594® donkey anti-rabbit IgG H&L (Cat. No. 711-585-152; Jackson ImmunoResearch, UK; diluted 1:500). After incubation, slides were washed in $3 \times 5$ min in TBST and mounted with Vectashield® mounting medium (Cat. No. H-1000; Vector Laboratories, USA).

All 2D sections were scanned using the automated Axio Scan.Z1 Slide Scanner (ZEISS, Germany), equipped with a Colibri 5/7 light source. DAPI, INS⁺ and GCG⁺ cells were imaged using an Axiocam 506 microscope camera (ZEISS, Germany) with a Plan-Apochromat 20×/0.8 M27 objective and the following filters: DAPI, 90 HE DAPI (Ex 353, Em 465); INS, 90 HE GFP (Ex 493, Em 517); and GCG, 64 HE mPlum (Ex 590, Em 618). For each islet, 3 Z-slices were taken at a range of 7 μm, with scans being saved in *.czi format and analyzed using ZEN (blue edition) microscopy software (version 3.7.97; ZEISS, Germany). To display GCG⁻ islets (Supplementary Fig. S14), ROIs were selected in ZEN and Z-stack images presented as orthogonal projections (maximum) with deblurring (strength, 0.5; BlurRadius, 15; and sharpness, 0).

### Statistics and reproducibility

Statistical data for surfaces generated using Imaris software from OPT and LSFM scans, including volumes, sphericity and overlap ratio between surface volumes, were exported as Excel (Microsoft Office 365, version 2304) *.xml files for quantification. For a given object, a 3D diameter was calculated as an average of the diameters in X, Y and Z axes. Graphs were generated and statistical tests performed using GraphPad Prism software (version 10.0.2; GraphPad Software, USA). Where appropriate, when selecting statistical tests, a Shapiro−Wilk test for normality was applied for each column. If the normality (gaussian distribution) test was passed, column data were subjected to a paired $t$-test for comparison (e.g., see Supplementary Fig. S9D, E and Fig. 4F) otherwise a Wilcoxon rank test was used (Fig. 4G). $P$-values are reported as: $P > 0.05$ (ns), $P \leq 0.05$ (*), $P \leq 0.01$ (**), $P \leq 0.001$ (***), $P \leq 0.0001$ (****).

### Reporting summary

Further information on research design is available in the Nature Portfolio Reporting Summary linked to this article.

## Data availability

Due to the large size of the raw and processed imaging datasets acquired by NIR-OPT and LSFM, these are available from the corresponding author. Data on mouse pancreata was downloaded from Dryad, open access public data repository: https://doi.org/10.5061/dryad.51c59zw8g (Dataset 1) (23). Source data are provided with this paper.

## Code availability

Scripts used for processing OPT data including alignment of axis of rotation post-OPT scanning (DFTA) (47) and Contrast Limited Adaptive Histogram Equalization (CLAHE) (20) are available as a compiled software package (together with video instructions on their implementation) at GitHub, https://github.com/ARDISDataset/DSPOPT.

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

## Acknowledgements

The authors thank Mrs. S. Ingvast, Mr. F. Morini and Mr. A. Ahlgren for initial technical assistance and Dr. S.M.A. Willekens for helpful comments on the manuscript. This work was supported by the Kempe Foundations (SMK-1455 to U.A.), the Swedish Research Council (2017-01307 to U.A. and 2023-02221 to O.K), Umeå University Biotech Grants (FS 2.1.6-2026-20 to U.A.), the Swedish Childhood Diabetes Foundation (Barndiabetesfonden) (to U.A. and O.K.), The Novo Nordisk Foundation (NNF21OC0069771 and 0084520 to U.A. and NNF20OC0063600 to O.K.), the Diabetes Wellness Foundation Sweden (PG21-6566 to U.A.), the Ernfors Family Fund (2023 to O.K.), Nils Eric Holmstens forskningsstiftelse (2023 to O.K.) and Diabetesfonden (DIA2021-59 to O.K.).

## Author contributions

J.L. performed OPT, LSFM and Axioscan imaging, performed statistical analyzes, was involved in study design and the development of the computational processing pipeline. W.D. was involved in study design and performed whole mount 3D immunohistochemistry. M.H. was involved in study design and initial development of the computational processing pipeline. O.K. conceived the study together with U.A., was involved in study design and responsible for sample collection. T.A. was involved in study design and performed 2D immunohistochemistry. U.A. conceived the study and supervised the entirety of the project. All

authors contributed to the interpretation of data, manuscript writing and approved the final version of the manuscript.

## Funding

## Competing interests
The authors declare no competing interests.
