## [Peer Review File · Nature Communications]

Illuminating the complete β -cell mass of the human pancreas -
signifying a new view on the islets of LangerhansEditorial Note: Parts of this Peer Review File have been redacted as indicated to remove third-party material where no permission to publish could be obtained.

REVIEWER COMMENTS

Reviewer #1 (Remarks to the Author):

In this manuscript, Lehrstrand et al. are presenting an imaging and analysis pipeline (based on a custom built near infrared Optical Projection Tomography microscope) to reconstruct the complete map of the beta cell mass in a whole human pancreas. Eventually, by simultaneously staining insulin and glucagon the authors come to the important and noteworthy realization that about 50% of the human islets are deprived of glucagon secreting cells. Also of note, they manage to discuss the differentiation of glucagon positive islets in terms of size, by showing that most of the gcg+ positive islets are generally among the biggest, while smaller islets are the ones without gcg producing alpha cells. This finding can open interesting question in the field of diabetes research, in particular regarding functionality of the islets in terms of size and metabolic impact, paracrine signaling concerning glucagon and insulin interplay. The imaging pipeline was previously published by the same group (Hahn et al. Communications biology 2021), but in this report, the optical sectioning has been improved to slices of 2.5 mm thickness. The methods are for the most part clearly written and it seems that the pipeline could be potentially reproduced. From the 5 donated pancreata the authors were able to analyze, 80% are coherent in terms of analyzed numbers, and one is slightly different, but still keeping the trend of the overall analysis. Although inherently limited by the number of donors, the claims of the paper seem to be well supported by the statistical analysis of the data acquired. Unfortunately, there's a number of misleading points concerning the correlation between the images and the quantification by numbers in the analysis. This is particularly critical since most of the important findings are linked to dimension, size and fraction of volume of insulin positive cells compared to total volume or total number of islets. These issues are the reason why a careful review is required before considering the publication of this article.

Major points to be addressed:

- In the introduction, the authors state “islet mass is suggested to be both unevenly and evenly distributed within the pancreas”. Looking at fig. 1 this does not seem to be the case, since it appears that INS+ objects are everywhere in the pancreas completely evenly distributed. I think more discussion on this point is needed and a little comment on how the authors position themselves in this debate, based on the images provided in this article.
- During the first paragraph of the results, the authors state that their OPT scanning results in an isotropic resolution of 21 microns. It could be nice to discuss more in details how this resolution is enough to clear the statements made in the manuscript, since one of the smallest volumes considered for islets has a reference diameter from 0 to 36 microns. 21 microns is way too big to consider diameters from 0 to at least 21 microns, so I would not add diameters or volumes smaller than the resolution achievable. This might lead to misleading arguments in the analysis.
- In the second paragraph of the results, the authors state that “the beta cell density is relatively uniform across the length of the pancreas, although variations exist among individual discs”, possibly due to differences in vascular and ductal densities. I think this point needs to be discussed much further in detail, since one of the major points of the paper is imaging a whole organ with all its spatial context available. Do the authors expect more presence of beta cells content if more vessels are present? Could it be possible to include an evaluation of the position of INS+ and GCG+ content in relation to the distance to blood vessels? Furthermore, could the author comment more in detail during the methodology section on how the fixation and the various cycles of hydration and dehydration could potentially affect the size/the space occupied by vascular and ductal densities?
- In the third paragraph of the results section, the authors state that smaller islets are more susceptible to autoimmune destruction. Since smaller islets are the ones lacking GCG+ cells, it will be interesting to comment more in detail and try to speculate a hypothesis regarding the two findings, otherwise people reading this paper might think that smaller islets are susceptible to autoimmune destruction just because they have no glucagon producing cells. Which might even be possible. I don’t know. I’m just saying it could be critical to add this statement like this at this point.
- The authors at one point, state that “the average islet size is significantly smaller than what reported by 2D studies” Since the distribution in terms of islet size looks pretty much gaussian, it will be important to add more information to this statement, maybe inserting some references and discussing also orders of magnitude.
- The main finding of the paper is the striking 50% of islets in the human pancreas which appear to be completely devoid of GCG+ cells. Have you tried to assess gene expression or have you done some sequencing, to see if these beta cell clusters show a different expression in terms of genes compared to the other islets containing both beta and alpha cells? Have you thought about what happens with somatostatin secreting delta cells? Probably they’re too small in number to be seen with the resolution of the OPT

- One of the key issues of this protocol that hasn't been discussed thoroughly in the paper, is how time consuming it is to collect all the data and manually align them for the 3D reconstruction. Also, it would be very interesting to address the size of the data collected, since storage of the data and computing power are nowadays the bottleneck for 3D optical microscopy.
- The data in figure 2 open a variety of concerns:
 - 1- Panel B shows INS+ volume contribution to total tissue volume per region, which is about 2,5%. How can it be 2,5% of the volume, if in the rendered volume in extended video and also in figure 1, the pancreas seems to be completely red (i.e. basically completely INS+)? This is very misleading, to me.
 - 2- As I said before, the resolution does not allow discerning objects smaller than 21 microns, so this should be the minimum value for the smaller diameter for the islets. (Panel C)
 - 3- In Panel F there's the plot of number of INS+ objects per mm². I think this number is way too small considering the pancreas depicted in fig.1 and in the extended dataset. Combining all the numbers, it seems there are about 60 islets of any size per mm². Maybe I'm mistaken, but the number seems too small for a squared millimeter.
Minor issue: The numbers in x axis are too small and too many to be read carefully.
 - 4- In the legend the authors talk about "empty space". It is critical to define better what they are talking about.
 - 5- The figure referred to can't be extended fig. 3 but more likely extended fig.4 this is important to be clarified for clarity of reading.
- Figure 3 panel I shows the average shortest distance among size categories. It is not clear to me how L-M and M-L are different qualitatively. I understand how they're considered, but in the end, isn't it the same measurement, made in 2 different perspectives? Can you please clarify this for me? Also, the colors of the bar graphs are very misleading, L-s and S-M are the same color but they are referring to different sizes. This is very misleading. What does the "All" bar graph mean, in this context? This is not very clear to me.
- For the mouse pancreas, the data show that the larger islets are located mostly on the "center axis". In this context, it is key to provide information on how the mice pancreata were collected, since it's not an isolated organ, as it is in humans, but it is more like a very "branched" and "liquid" piece of tissue. Depending on the collection, the islets might be located in different positions.
- Figure 4: I think scale bars need adjustments. I don't think scale bar in B can be 100 microns if in panel A is 200 microns. The size of the bigger object in the image is about 200 microns, but looking in panel B it looks at least 300-400 microns. This is crucial, since the biggest finding in the paper states that size matters for different islet organization.
- In extended fig. 2 it is stated that the segmentation accuracy is 102%. How's an accuracy be more than 100%? I understand it has been calculated as the ratio between manual

segmentation and pipeline segmentation, but 102% still does not mean anything. I would re-arrange the figure, by excluding the manual annotation (useless, in these days in which there's a plethora of segmentation softwares, based also on machine/deep learning, which are very accurate) and I would also re-arrange the extended data table 2 accordingly. 119% accuracy of segmentation is not something reasonable to put on a scientific paper, in my personal opinion. It leads to misdirection such as 19% of additional artifacts, for example? This needs to be addressed carefully.

- I don't personally see the difference between Extended data figure 3 and fig.1D. However, this image led me to realize that measuring the pancreas with that scale bar yields a length of about 14 cm, while 51 slabs of 2.5 mm yield a length of 12,75 cm. What happened to the remaining cm? Is this loss of volume part of the fixation protocol? This needs to be addressed very carefully since the whole organ imaging is one of the major key aspects of the paper.

Minor points:

- Images in the figures and in the extended figures are often displayed in green/red and sometimes even in red with black background. People with color blindness sometimes are lacking the capability to detect the lower color wave frequencies associated with red, thus confusing it with black. The red on black images will not be able to be recognized as such. I suggest a different look up table for the images. Maybe based on the CMYK color model.
- In the introduction, the authors talk about a "previously unknown heterogeneity in islet composition", although there are a number of reports (some of them cited and referenced by the authors themselves) which are addressing this issue. To be noted, this heterogeneity has been studied also during development (see recent work by Sasaki et al. *Diabetes Metab J* 2023;47:173-184 and references within, Glorieux et al, 2022 *Development* and references within, and also Miranda et al. *Am J Physiol Endocrinol Metab*320: E716–E731, 2021). So I think some comment should be added on this, and eventually add the references as well. If the heterogeneity they discuss about refers only to introducing the lack of GCG producing cells in 50% of the islets, they should be clearer in stating about heterogeneity of islet composition.
- Since it is a diabetes-related article, I would suggest a reference for ultramicroscopy for the general audience, even if it's nowadays a pretty much established technique. Maybe the seminal work by Voie et al. *Journal of Microscopy* 1993 could be cited, for the general public.
- In the third paragraph of the results section, "optical" needs to be replaced by "optically", "between" with "among".
- I would discuss a little bit more in detail about the resolution limitation during the discussion in which the authors talk about the possibility "to study for the first time from all angles and through its entire depth"
- It is true that the developed approach overcomes the reagent penetration issue, but what about discussing a little more about the potentially dangerous effects of the

various cycle of dehydration/hydration, and of the clearing procedure on tissue shrinkage, which will hinder morphological quantification?

- In the methodology section I missed how the mice organs were collected. It is important to add it, since the mice pancreata are generally not very precise in shape, and the methodology of the collection of the sample needs to be addressed to have full reproducibility of the protocol.
- Extended data fig, 10 please fix the scale bars there are too many and in different shapes and sizes.
- Extended data fig. 12. Aare you sure the scale bar is 800 microns? From similar images in extended data fig.13 it looks like the sizes are way different.

Reviewer #2 (Remarks to the Author):

Lehstrand et al

The authors present a technical tour de force in quantifying the number, size and cellularity of islets across the entirety of five whole pancreata from adult human subjects, using optical projection tomography and image reconstruction. The really important advance versus earlier studies is the projection of 3D images across the entire pancreas.

Key observations are that:

1. The average size of human islets is smaller than previously assumed
2. 25 % of islets contain 75 % of the beta cell mass
3. Confirmation of findings from Farhat et al (2013) that smaller islets have more INS+, GCG-cells
4. A substantial fraction of (mostly small) islets have no detectable GCG+ cells
5. A generally similar distribution of islet density, sizes and cellular composition is observed across the whole pancreas

Major

1. Whilst I congratulate the authors on a terrific technical achievement, there is a sense of “overselling” of some of the data, with the repeated emphasis on the result (e.g. line 166) that “..the majority of islets lack GCG+ cells”. This is true, of course, but is skewed by the fact that this observation chiefly refers to smaller islets (Fig 4H and extended data Fig 7E – though I think INS+GLU- cells are mislabelled?). The better way of looking at this is to ask how many beta cells belong to an islet without GCG+ cells? At this point, I would guess that it is the minority! Indeed, this view is supported by the data in Fig 4G (and extended data Fig 7E – though I think INS+GLU- cells are mislabelled?) wherein only about 15 % of the total islet volume comprises GCG- islets. The degree to which insulin secretion is regulated by locally released glucagon (or indeed GLP1) is still a matter of debate. I think this finding needs to be toned down.
2. Fig. 2 It be helpful to show summary data for all five pancreata analysed. We are left without a sense of variation in in islet size, cellularity etc. between donors.
3. I wasn't always convinced that the authors provide the most straightforward explanation of their findings, e.g. lines 211-213. Surely a paucity of smaller islets after islet pancreatic disruption and isolation is likely to be due – at least in part - to their loss by digestion/mechanical damage?

4. How do the authors explain the differences in islet distribution (more islets in the tail in the studies from Wang et al, 2013?) Could there be differences in the subjects used (age, MBI, ethnicity) or other technical issues (pancreas and data treatment)?
5. I was missing the comparison to mouse pancreata in terms of the proportion of islets that are GCG-. Presumably published in an earlier report from this lab?
6. I wonder why other islet cell types e.g. Sst+ were not measured?
7. The study would be significantly reinforced by examination of pancreata from subjects with type 1 or type 2 diabetes.

Reviewer #3 (Remarks to the Author):

Professor Ahlgren has been pioneering optical projection tomography imaging of the pancreas, with studies providing valuable insights into changes occurring to beta cells - both in T1D and T2D. While most of his laboratory's studies were performed in rodents, a few years ago they provided the first complete OPT analysis of the human pancreas. They showed the 3D distribution of islets within human pancreata from non-diabetic and T2D donors. In the present manuscript they extend their studies by documenting the distribution of alpha cells and their co-presence with beta cells in individual islets throughout the entire human pancreas. One of the main findings is that a relatively large number of islets are devoid of alpha cells, which has important implications in terms of biological understanding of endocrine pancreas biology and further sheds light on islet heterogeneity. This is a significant finding that requires to be strongly supported by the author's data and analysis. In this respect my main questions are:

- The identification and quantifications of ROIs are highly dependent on intensity thresholds used during image processing. This could artificially lead to an underestimation of the alpha cell population for instance, which would have important consequences on the conclusions of this study. How did the authors confirm that thresholds used for both insulin and glucagon signals are reflecting the real number and ratio of these cells within the human pancreas?
- Please clarify how many samples were used for all data analysis. Also, it is unclear from the abstract and introduction how many pancreata were assessed for this manuscript.

In addition I would like the authors to address these comments:

- It is stated that this manuscript provides the first complete representation of beta cells throughout the human pancreas (for example lines 20, 66, 204), however this has been done in one of their previously published studies.
- The statement in the abstract, “50% of the human insulin-expressing islets are virtually devoid of glucagon-producing alpha cells” is misleading. As we understand later in the text, 16% of BCM is comprised of islets devoid of alpha cells. This is a much lower number and the authors should discuss the relevance of this finding for the entire endocrine pancreas function.
- The authors previously assessed the beta cell mass and distribution in a human T2D pancreas, and correctly mention in the introduction that “diabetes is a disease that involves all islet cell types”. Did the authors assess alpha cell distribution in a human T2D pancreas and/or could discuss how their current finding could apply to what happens in T2D?
- From their previously published study, it was found that there are higher islet density areas in the organ periphery. Was this confirmed in the present study?
- The authors should include a comparison of their results with previous published studies assessing beta cell mass in the human pancreas (using other methodologies, such as histology or flow cytometry). Also, findings regarding the ratio of alpha/beta cells should be compared to the general statement that beta cell content in human islets represents approximately 40-60% of all islet endocrine cells (see for example the review from Campbell et al., 2021, Nat Rev Mol Cell Biol).
- In the Materials and Methods part, experiments on mouse pancreas are missing.
- Figure 2: the average INS+ volume / anatomy volume is 3%, this should be compared to the literature. For consistency in the headings, the authors should decide between “volume”, “vol.”, “vol”. Panels A and B, Region 1: there are more data points presented in B than in A, could you explain what the individual data points represent? Finally, in panel A, although this is explained in the text, the outlier bar should be marked differently (with “*” reserved for statistics). The bar itself could be distinguished as outlier by a different color and/or pattern.
- Figure 3: was one human pancreas analysed in panel B, and five in panels H and I? Is the analysis on mouse pancreas based on 5 samples for D, for comparison with B? Since this

panel compares distribution and sizes of islets in human versus mouse, the same information should be given for both species: panels E-I should be followed by similar panels for mouse. This would support the statement on lines 656-658.

- Figure 4, panel F, what does each individual data point represent? From this panel, about 15% of islets contain alpha cells and are devoid of beta cells, is this correct?
- Figure 4, panel H, data should be clearly labeled as in panels F and G (and include INS-GCG+). Also, when looking at the size category 50-100 in Fig 2F, there are a total of approx. 8 insulin positive objects per mm³, contrasting with a total of about 20 objects in Fig 4H at the same size category, could you explain this discrepancy?
- Figure 4, panel J, the average islet content in alpha cells is 10-20% in the present study, which is low as compared to previous published studies (again, see for example Campbell et al., 2021, Nat Rev Mol Cell Biol). Can this be explained?
- Concerning the average alpha:beta cell ratio, a combined value is presented in Figure 4. Ratios for each individual donor should be provided in addition to this combined value. Ideally, ratios should also be provided individually per size category.

Finally, some minor points:

- For consistency, the authors should use beta cell mass (BCM) and not BCV throughout the text (for example lines 85, 114, 126, 138).
- Line 38, “[...] islet volume of 0.5-2.0 cm³.”, please provide reference.
- Line 82 and elsewhere, “and Fig. 3” is confusing as it appears to direct to Fig. 3 and not to the supplementary figure.
- Line 118, could you confirm the this is “average” and not “median”?
- Line 119: the correction for tissue processing is confusing here. Is it corrected elsewhere or are all other values, including in the figures, left uncorrected? How the corrected value was calculated should also be explained or a reference provided.
- Line 146, “(see above)”, which other reports are meant? Should it be a reference?
- Lines 151-153, “[...] and in contrast [...]”, it is not clear what the authors mean.
- Lines 159-161, “the average islet size is significantly smaller than what has been reported by 2D stereological studies”, please provide data and references.
- Lines 161-162, “human beta cell mass organization differs significantly from that of the

mouse”, please provide data to support this conclusion.

- Line 175, “consensus 2:1 beta cell to alpha cell ratio”, and line 220, please provide reference(s).
- Line 249, “small islets have superior cellular function compared with larger islets”, the reference 30 regards islet transplantation and not islets in situ.
- Line 251, small mouse islets have been reported to be almost devoid of alpha cells. This might be due to the isolation procedure and not a confirmation of the present findings (furthermore, mouse islets have a much lower percentage of alpha cells).
- Figure 1D, and Extended Data Fig. 3 are identical and almost of same size, why is there an Extended Data Fig. 3?
- Extended Data Fig. 6, panel D: large islets seem to be more present in the periphery of the sample, is this due to the sample processing/staining procedure? Legend for panels Q-R, images are representative of how many mouse pancreas samples?
- Extended Data Fig. 11, all samples should be presented individually, rather than having combined values for H2456, 2457, 2466, 2522.
- Extended Data Fig. 12 is not referred to in the main text.

Despite all of the above, this is an impressive study providing essential information on the composition of the human pancreas at high resolution. I am looking forward to the revised manuscript.

Response to referees Lehrstrand et al.,

We would like thank the reviewers for their constructive comments, which we feel has contributed to further strengthen our manuscript. We have addressed their comments in a point-by-point list below (in blue). We are submitting two versions of the manuscript and the supplementary info, one clean and one in which changes are outlined in red, labelled markup.

REVIEWER COMMENTS

Reviewer #1 (Remarks to the Author):

In this manuscript, Lehrstrand et al. are presenting an imaging and analysis pipeline (based on a custom built near infrared Optical Projection Tomography microscope) to reconstruct the complete map of the beta cell mass in a whole human pancreas. Eventually, by simultaneously staining insulin and glucagon the authors come to the important and noteworthy realization that about 50% of the human islets are deprived of glucagon secreting cells. Also of note, they manage to discuss the differentiation of glucagon positive islets in terms of size, by showing that most of the gcg+ positive islets are generally among the biggest, while smaller islets are the ones without gcg producing alpha cells. This finding can open interesting question in the field of diabetes research, in particular regarding functionality of the islets in terms of size and metabolic impact, paracrine signaling concerning glucagon and insulin interplay. The imaging pipeline was previously published by the same group (Hahn et al. Communications biology 2021), but in this report, the optical sectioning has been improved to slices of 2.5 mm thickness. The methods are for the most part clearly written and it seems that the pipeline could be potentially reproduced. From the 5 donated pancreata the authors were able to analyze, 80% are coherent in terms of analyzed numbers, and one is slightly different, but still keeping the trend of the overall analysis. Although inherently limited by the number of donors, the claims of the paper seem to be well supported by the statistical analysis of the data acquired. Unfortunately, there's a number of misleading points concerning the correlation between the images and the quantification by numbers in the analysis. This is particularly critical since most of the important findings are linked to dimension, size and fraction of volume of insulin positive cells compared to total volume or total number of islets. These issues are the reason why a careful review is required before considering the publication of this article.

Major points to be addressed:

R1-1.- In the introduction, the authors state “islet mass is suggested to be both unevenly and evenly distributed within the pancreas”. Looking at fig. 1 this does not seem to be the case, since it appears that INS+ objects are everywhere in the pancreas completely evenly distributed. I think more discussion on this point is

needed and a little comment on how the authors position themselves in this debate, based on the images provided in this article.

As noted below (see answer to question regarding Fig. 2, R1-10) it is hard to make any assumptions on BCM distribution based on a Maximum intensity projection (MIP). However, our quantitative data provide a more detailed view of BCM distribution. As the reviewer highlights, our data does not support the case where there is a higher islet density in the pancreatic tail based on insulin expression when analysing an entire pancreas, although variations exist between individual discs. Further, analyses of individual discs from regions 1-4 in other pancreata (See **New Fig. S4** showing; A, the INS⁺ volume/anatomy volume for individual discs from region 1-4 and, B/ INS⁺ volume/anatomy volume per size category in all 5 investigated pancreata) does not lay bare an obvious trend towards increased BCM in the tail. We have in the revised manuscript made a note on this matter in the discussion (line 225 and forward). Of note, in other studies, by e.g., in Ionescu-Tirgoviste et al., Sci Reports, 2015, islet area was estimated (based on H/E) staining's) on 5423 islets in total whereas in the study by Wang. et al., PLoS One, 2013, on 2D data obtained from 5µm paraffin sections, the area and frequency of four hormonal cell types were investigated. In the latter study, it is unclear to us exactly how many islets were investigated and the sampling frequency. In our study, on the other hand, we have analysed the 3D volume of INS⁺ cells in 2.21x10⁶ islets in one entire pancreas and in 4 discs from region 1-4 from four additional donor pancreata, each disc encompassing around 30 000 INS⁺ islets.

Finally, it should be noted that the regions of the human pancreas are quite loosely defined into head, neck, body, tail (in particular between the latter two). It is, therefore, often unclear how these regions have been defined in material from other studies.

R1-2. During the first paragraph of the results, the authors state that their OPT scanning results in an isotropic resolution of 21 microns. It could be nice to discuss more in details how this resolution is enough to clear the statements made in the manuscript, since one of the smallest volumes considered for islets has a reference diameter from 0 to 36 microns. 21 microns is way too big to consider diameters from 0 to at least 21 microns, so I would not add diameters or volumes smaller than the resolution achievable. This might lead to misleading arguments in the analysis.

We acknowledge the concern of the reviewer and this limitation in resolution has now been emphasized in the relevant figure legends. "Note, the resolution limit of the scanner at the implemented zoom is 21µm.

R1-3.- In the second paragraph of the results, the authors state that "the beta cell density is relatively uniform across the length of the pancreas, although variations exist among individual discs", possibly due to differences in vascular and ductal densities. I think this point needs to be discussed much further in detail, since one of the major points of the paper is imaging a whole organ with all its spatial context available.

We appreciate the comment by the reviewer but are not exactly sure what the reviewers wants us to discuss. As stated in the manuscript "... whereas the walls of vessels and ducts are included into the total tissue volume, "empty space" contained within these structures are not". Hence, the sentence merely suggests that more or

less vascular and/or ductal tissue could be present in certain regions of the organ, which may contribute to this variation since these tissues are included in the “anatomy volume” towards which the INS⁺ volume is normalized. Please see also the answer to the question below and to R1-14.

R1-4. Do the authors expect more presence of beta cells content if more vessels are present? Could it be possible to include an evaluation of the position of INS⁺ and GCG⁺ content in relation to the distance to blood vessels?

This is indeed a possibility which however requires a careful 3D reconstruction of the vascular system in relation to the segmented islet tissue to fully benefit from 3D image analysis. Since addressing this issue was not the primary aim of this investigation, suitable vascular markers were not included in the whole mount immunohistochemical design. In future studies, partly initiated, we plan to incorporate such markers as well as marker for innervation to address the possibility that islets of different cellular make up are differently vascularised/innervated. Of note, we are currently exploring AI tools to segment the vascular and ductal system based on their endogenous autofluorescent properties. This however, although it may help resolve the above issue should in our opinion be considered as a study by its own right. There are of course numerous questions regarding the relationship between different pancreatic constituents, both on a molecular and cellular level, and the current study will serve as a foundation to address several of these.

R1-5. Furthermore, could the author comment more in detail during the methodology section on how the fixation and the various cycles of hydration and dehydration could potentially affect the size/the space occupied by vascular and ductal densities?

A number of tissue processing/clearing protocols are currently in use for optical 3D imaging, including solvents such as e.g., BABB and different DISCO variants. Most of these, to various degrees, influence tissue size (for organic solvents normally a shrinkage effect and for aqueous based solvents normally an expansion effect, for review see e.g., Richardson and Lichtman, Cell, 2015). We assume that the reviewer is asking whether or not cycles of hydration and dehydration potentially affects shrinkage of vascular and ductal tissues differently from islets and exocrine tissues, and therefore would have an effect on normalisation when calculating islet density. In most literature, a general shrinkage effect using the applied dehydration and clearing protocols is in the range of 10-15% and it has been suggested that shrinkage is less in larger tissue specimen (see e.g., Vulderson et al., PLoS One 2021 and Laboratory Histopathology. A complete reference. Edition: First Chapter: Section 4 Tissue and section preparation. Publisher: Churchill-Livingstone, Editors: A.E. Woods & R.C. Ellis, 1994). We have in the revised manuscript included a new analysis illustrating the degree of tissue shrinkage in the current material. This analysis shows a tissue shrinkage effect of ~5% measured on entire pancreatic discs (see **new Fig. S7**).

R1-6. - In the third paragraph of the results section, the authors state that smaller islets are more susceptible to autoimmune destruction. Since smaller islets are the ones lacking GCG⁺ cells, it will be interesting to comment more in detail and try to speculate a hypothesis regarding the two findings, otherwise people reading this paper might think that smaller islets are susceptible to autoimmune destruction just

because they have no glucagon producing cells. Which might even be possible. I don't know. I'm just saying it could be critical to add this statement like this at this point.

The statement was given as an example of one type of analysis previously performed by OPT imaging in rodents that would have a clear added value if possible to perform also in human pancreas (see line 137-143 and onwards). Without having performed such analyses in human T1D donor material (which is very difficult to obtain in large volumes required for the type of study we present here), we feel that proposing a valid data-driven hypothesis linking smaller islets lacking glucagon with any autoimmune disease would be seen as too speculative. As a note to the reviewer, we are currently using the described approach to invest β -cell decay in material from a few individuals with long standing T1D and what he/she is suggesting is definitively something worth looking into in future studies if relevant material can be obtained.

R1-7. - The authors at one point, state that “the average islet size is significantly smaller than what reported by 2D studies” Since the distribution in terms of islet size looks pretty much gaussian, it will be important to add more information to this statement, maybe inserting some references and discussing also orders of magnitude.

We have in the revised manuscript added references to other average islet size measurements (see lines; 37-38), and further made a comment about this issue in the discussion section, (see lines 240-250).

R1-8. - The main finding of the paper is the striking 50% of islets in the human pancreas which appear to be completely devoid of GCG+ cells. Have you tried to assess gene expression or have you done some sequencing, to see if these beta cell clusters show a different expression in terms of genes compared to the other islets containing both beta and alpha cells? Have you thought about what happens with somatostatin secreting delta cells? Probably they're too small in number to be seen with the resolution of the OPT.

This is a justified question: we understand it is not something that is being asked for by the reviewer to be incorporated in this current study. Indeed, our future research endeavours will focus on understanding the significance of the islet's compositional heterogeneity on islet function by a range of techniques.

R1-9. - One of the key issues of this protocol that hasn't been discussed thoroughly in the paper, is how time consuming it is to collect all the data and manually align them for the 3D reconstruction. Also, it would be very interesting to address the size of the data collected, since storage of the data and computing power are nowadays the bottleneck for 3D optical microscopy.

Indications of typical time consumption and required data storage is now introduced in the material and methods section of the supplementary information under a separate subheading entitled “Data size and time consumption”.

R1-10. - The data in figure 2 open a variety of concerns:

1- Panel B shows INS+ volume contribution to total tissue volume per region, which is about 2,5%. How can it be 2,5% of the volume, if in the rendered volume in extended video and also in figure 1, the pancreas seems to be completely red (i.e. basically completely INS+)? This is very misleading, to me.

Here we would like to point out that the presented images of the entire pancreas (Fig 1D, Movie S2 and Figure S3 (a higher resolution image of 1D)) are, as stated in the text, presented as maximum intensity projection (MIP) images. I.e., we are in these seeing all labelled objects at all levels at the same time, also those that are lying “on top of each other”, which generates this optic effect. We are in fact simultaneously displaying 2.21 million objects in about 45 cm³ of tissue. Thus, the fact that the labelled cells only constitute a fraction of the entire pancreatic volume (or 2.8%, see figure legend) is perhaps best illustrated by looking at tomographic slices of the tissue (see movie S1 starting at e.g., 13 s and beyond).

R1-11. 2- As I said before, the resolution does not allow discerning objects smaller than 21 microns, so this should be the minimum value for the smaller diameter for the islets. (Panel C)

Please see the response provided above, **R1-2**, that addresses this question.

R1-12. 3- In Panel F there's the plot of number of INS+ objects per mm². I think this number is way too small considering the pancreas depicted in fig.1 and in the extended dataset. Combining all the numbers, it seems there are about 60 islets of any size per mm². Maybe I'm mistaken, but the number seems too small for a squared millimeter.

The total number of INS⁺ islets are 2.21×10^6 (or more precisely 2 206736) and the volume of the full pancreas is 45.188 cm³ or 45 188 mm³. The total number of islets divided by the entire tissue volume is $2206736/45.188 = 48.83$ islets/mm³. The other way around this means that there is on an average one islet per 0.02 mm³ of tissue. Of note (as mentioned in the manuscript), the displayed data show a normalized average islet/disc ratio to better illustrate the rather limited degree of variation in size categories in different regions of the pancreas as observed in our in-depth study. We have in the revised manuscript added data for the distribution of size categories calculated based also on the entire islet population normalized to the entire tissue volume (Grey lines, **New Figs. 2 D-G**).

R1-13 Minor issue: The numbers in x axis are too small and too many to be read carefully.

In order to provide comprehensive distribution statistics, we felt that the number of size categories was necessary. However, we are happy to change the images, that are zoomable, on the editor's discretion.

R1-14 4- In the legend the authors talk about “empty space”. It is critical to define better what they are talking about

The original sentence “Note, whereas the walls of vessels and ducts are included into the total tissue volume, “empty space” contained within these structures are not”

is now changed to:

“Note, whereas the walls of vessels and ducts are included in the total tissue volume, the lumens of these structures (i.e., empty space) are not (See fig. legend Fig. 2, lines 407-409)

R1-15. 5- The figure referred to can't be extended fig. 3 but more likely extended fig.4 this is important to be clarified for clarity of reading.

We thank the reviewer for noticing this. The reference to Fig. S3 was erroneous and it should be **Fig. S6** (in the revised manuscript). This has now been corrected.

R1-16. - Figure 3 panel I shows the average shortest distance among size categories. It is not clear to me how L-M and M-L are different qualitatively. I understand how they're considered, but in the end, isn't it the same measurement, made in 2 different perspectives? Can you please clarify this for me?

We feel the data provide a good representation on how islets of different size categories are distributed in relation to each other in 3D space. The values we calculate and include give a good idea about the distance between islets of different sizes (i.e., grouped into Small, Medium and Large), and these reflect the number of islets of each category within the pancreas. For example, the average distance from each large islet to medium sized islets (which are more numerous) is shorter than the other way around (see image below)

R1-17. Also, the colors of the bar graphs are very misleading, L-s and S-M are the same color but they are referring to different sizes. This is very misleading. What does the “All” bar graph mean, in this context? This is not very clear to me.

We thank the reviewer for pointing this out. The colour coding of S-L and S-M has switched place. This is now corrected in the revised manuscript. As for the “All” bar graph it simply denotes the average distance to the 5 nearest neighbors regardless of size category (see figure to the left). This has now been clarified in the figure legend.

R1-18. - For the mouse pancreas, the data show that the larger islets are located mostly on the “center axis”. In this context, it is key to provide information on how the mice pancreata were collected, since it’s not an isolated organ, as it is in humans, but it is more like a very “branched” and “liquid” piece of tissue. Depending on the collection, the islets might be located in different positions.

**TWO FIGURES
REDACTED**

We disagree with the reviewer in that the mouse pancreas would be a “liquid piece of tissue” and not an isolated organ. We have in several previous publications demonstrated that the mouse pancreas is a well-defined organ consisting of three clearly discernible primary lobes (dorsal or splenic, ventral or duodenal and gastric), which we advocate should be designated based on their developmental origins (see e.g., “The Pancreas” in Hörnblad et al., Kaufmans Atlas of Mouse Development 2016). Indeed, this raises questions also about the current nomenclature of the human pancreas which is not strictly based on anatomical/biological boundaries, but that is outside the scope of the current manuscript. The primary lobes of the mouse pancreas can easily be

identified and isolated individually (see e.g., Eriksson et al., Jove 2013 and Hahn & Ahlgren Methods Mol Biol, 2023). Hence, we disagree that the collection of tissue would in any way make the islets become repositioned. As for the actual data presented here, as stated in the manuscript, it is derived from another study (Hahn et al., Communications Biology 2020, Hahn. Et al., Scientific Data 2022) in which the

utilized collection procedure is clearly referenced. In the revised manuscript, adequate references to the mouse pancreas isolation and processing have been added in the supplementary information under the subheading “Assessment of insulin islet 3D distribution and organization”.

R1-19. - Figure 4: I think scale bars need adjustments. I don't think scale bar in B can be 100 microns if in panel A is 200 microns. The size of the bigger object in the image is about 200 microns, but looking in panel B it looks at least 300-400 microns. This is crucial, since the biggest finding in the paper states that size matters for different islet organization.

We are grateful to the reviewer for pointing this out. The scalebar in Fig. 4A is erroneously labelled, and should be 500 μ m instead of 200 μ m. This is now corrected in the figure legend.

R1-20. - In extended fig. 2 it is stated that the segmentation accuracy is 102%. How's an accuracy be more than 100%? I understand it has been calculated as the ratio between manual segmentation and pipeline segmentation, but 102% still does not mean anything. I would re-arrange the figure, by excluding the manual annotation (useless, in these days in which there's a plethora of segmentation softwares, based also on machine/deep learning, which are very accurate) and I would also re-arrange the extended data table 2 accordingly. 119% accuracy of segmentation is not something reasonable to put on a scientific paper, in my personal opinion. It leads to misdirection such as 19% of additional artifacts, for example? This needs to be addressed carefully.

We are not exactly sure what the reviewer means here. This figure and the accompanying supplementary table are provided to give an estimate of the accuracy of the segmentation routines used for the statistical analyses of OPT data in the manuscript. Indeed, there are a number of software for segmentation. Is the reviewer asking us to use other routines to evaluate the routines we have implemented in the manuscript, and in such case how do we evaluate these routines? The reviewer is correct in that an accuracy cannot be >100% and we agree that “accuracy” in this case is the wrong term to use for the presented analysis. In the revised manuscript we have therefore exchanged the term “accuracy” to “relative percentage difference” which is commonly used in quantitative science as an indicator of quality assurance where the outcomes are expected to be the same.

R1-21. - I don't personally see the difference between Extended data figure 3 and fig.1D.

They are indeed the same, but whereas Fig. 1D is included for illustration of the data generation process, Fig. S3 is included as a higher-resolution image of the complete BCM distribution of a human pancreas devoid of labelling (boxes, arrows, lettering) that would otherwise disturb the picture.

R1-22. However, this image led me to realize that measuring the pancreas with that scale bar yields a length of about 14 cm, while 51 slabs of 2.5 mm yield a length of 12,75 cm. What happened to the remaining cm? Is this loss of volume part of the

fixation protocol? This needs to be addressed very carefully since the whole organ imaging is one of the major key aspects of the paper.

As can be seen in Fig. S1A, the pancreas is embedded in agarose in a 3D printed matrix to facilitate cutting of the organ into discs of the same width. As measured in the matrix, the pancreas is 14 cm. The cavities between the teeth of the matrix are 0.3mm, which is required to allow room for the pathology dissection blade when slicing the tissue. Hence, the slices become slightly bigger than 2.5mm (approximately 0.15 mm on each side, i.e., 2.8 mm). I.e., approximately $50 \times 0.3 = 1.5$ cm which constitutes the “missing tissue”. When writing the manuscript, we erroneously indicated the “teeth” size, not including the addition of the cavities between them. This is now corrected in the revised manuscript (see new **Fig. S1**, figure legend and first sentence in the methods section under “Tissue preparation for 3D imaging”, “A slicing matrix was designed with a section thickness of 2.8 mm, providing an optimal balance between reagent penetration.....” Of note, slight morphological changes of the tissue discs during tissue processing and labelling may further contribute as this sometimes prevents perfect alignments between discs (E.g., if one disc becomes slightly curved). This however does not influence quantitative assessments of islet or tissue volumes.

Minor points:

R1-23. - Images in the figures and in the extended figures are often displayed in green/red and sometimes even in red with black background. People with color blindness sometimes are lacking the capability to detect the lower color wave frequencies associated with red, thus confusing it with black. The red on black images will not be able to be recognized as such. I suggest a different look up table for the images. Maybe based on the CMYK color model.

We have tried to follow a colour scheme that could be considered more or less a consensus in the field to make our 3D images easier to relate to previous 2D fluorescent studies (google “islets of Langerhans + glucagon + insulin + fluorescence”). This further has the benefit of clearly showing co-expression if relevant exist (i.e., red + green = yellow). We are however aware of the problem and are of course willing to change the colour scheme on the editor’s discretion.

R1-24. - In the introduction, the authors talk about a “previously unknown heterogeneity in islet composition”, although there are a number of reports (some of them cited and referenced by the authors themselves) which are addressing this issue. To be noted, this heterogeneity has been studied also during development (see recent work by Sasaki et al. *Diabetes Metab J* 2023;47:173-184 and references within, Glorieux et al, 2022 *Development* and references within, and also Miranda et al. *Am J Physiol Endocrinol Metab*320: E716–E731, 2021). So I think some comment should be added on this, and eventually add the references as well. If the heterogeneity they discuss about refers only to introducing the lack of GCG producing cells in 50% of the islets, they should be clearer in stating about heterogeneity of islet composition.

We agree with the reviewer and although some of these references were already in the manuscript, we have clarified this point and added relevant additional references (see lines 60-61).

R1-25. - Since it is a diabetes-related article, I would suggest a reference for ultramicroscopy for the general audience, even if it's nowadays a pretty much established technique. Maybe the seminal work by Voie et al. Journal of Microscopy 1993 could be cited, for the general public.

In the revised manuscript adequate references are included as suggested (see line 80).

R1-26. - In the third paragraph of the results section, “optical” needs to be replaced by “optically”, “between” with “among”.

The first sentence “As demonstrated optical by 3D imaging” had a typo and was supposed to read “As demonstrated by optical 3D imaging”. This has now been corrected.

As for the use of “between” vs “among”, <https://www.grammarly.com/blog/between-among/> states “.. you can use *between* for any number of elements, as long as all the elements are separate and distinct. According to *The Chicago Manual of Style*, you can even use *between* when “multiple one-to-one relationships are understood from the context”. As stated above (see answer to R1-18 above), the elements referred to (the primary lobes of the mouse pancreas) are separate and distinct.

R1-27. - I would discuss a little bit more in detail about the resolution limitation during the discussion in which the authors talk about the possibility “to study for the first time from all angles and through its entire depth”

We are not entirely sure what the reviewer asks for here. To the best of our knowledge, our study represents the first global analysis of the human pancreas using highly specific antibodies by optical imaging. This gives essentially full freedom of target selectivity, enabling assessments of 3D coordinates, volumes and shapes etc. for all labeled objects, at the current level of resolution (resolution mentioned multiple times in the manuscript). Indeed, nuclear and radiologic imaging techniques also allows the pancreas to be studied from all angles and through its entire depth, but these are generally greatly limited in the available pool of specific contrast agents (e.g., CT/MRI) and/or resolution (e.g., PET/SPECT). The sentence is intended to be read in context of the possibility to use specific antibody labelling (or potentially other fluorescent markers) in combination with optical 3D imaging. To avoid confusion the sentence “*The developed possibility to study the pancreas for the first time from all angles and through its entire depth (with known 3D coordinates, volumes and shapes for all labeled objects....)*” has in the revised manuscript been changed to “*The developed possibility to study the human pancreas using specific antibody labelling, from all angles and through its entire depth (delivering 3D coordinates, volumes and shapes for all labeled objects)....*” (See lines 314-316).

R1-28. - It is true that the developed approach overcomes the reagent penetration issue, but what about discussing a little more about the potentially dangerous effects of the various cycle of dehydration/hydration, and of the clearing procedure on tissue shrinkage, which will hinder morphological quantification?

Firstly, we feel that these procedures have been well described and validated in the references given in the manuscript (See e.g., Alanentalo et al., Nature Methods 2006 (showing that the reversal of the clearing protocol allows subsequent sectioning and staining with preserved morphology) and Hahn et al., Communications Biology 2021 (in which islet diameters calculated from 3D datasets generated by OPT and LSM was compared to paraffin sections (which are also subjected to dehydration) of the very same tissues (See Supplementary Fig 3 of Hahn et al., Communications Biology 2021). See also Hahn et al., Sci Reports 2020 for studies on human pancreas tissue. We are therefore confident that the data can be compared with this variation in mind. See also response to reviewer #3, R3-19, below)

R1-28. - In the methodology section I missed how the mice organs were collected. It is important to add it, since the mice pancreata are generally not very precise in shape, and the methodology of the collection of the sample needs to be addressed to have full reproducibility of the protocol.

See answer to R1-18, above.

R1-29. - Extended data fig, 10 please fix the scale bars there are too many and in different shapes and sizes.

This is now corrected in the revised manuscript (now Fig. S14).

R1-30. - Extended data fig. 12. Are you sure the scale bar is 800 microns? From similar images in extended data fig.13 it looks like the sizes are way different.

These are correct, please note that the images represent very different zoom factors.

Reviewer #2 (Remarks to the Author):

Lehstrand et al

The authors present a technical tour de force in quantifying the number, size and cellularity of islets across the entirety of five whole pancreata from adult human subjects, using optical projection tomography and image reconstruction. The really important advance versus earlier studies is the projection of 3D images across the entire pancreas.

Key observations are that:

1. The average size of human islets is smaller than previously assumed
2. 25 % of islets contain 75 % of the beta cell mass
3. Confirmation of findings from Farhat et al (2013) that smaller islets have more INS+, GCG- cells
4. A substantial fraction of (mostly small) islets have no detectable GCG+ cells
5. A generally similar distribution of islet density, sizes and cellular composition is observed across the whole pancreas

Major

R2-1. Whilst I congratulate the authors on a terrific technical achievement, there is a sense of “overselling” of some of the data, with the repeated emphasis on the result (e.g. line 166) that “..the majority of islets lack GCG+ cells”. This is true, of course,

but is skewed by the fact that this observation chiefly refers to smaller islets (Fig 4H and extended data Fig 7E – though I think INS+GLU- cells are mislabelled?). The better way of looking at this is to ask how many beta cells belong to an islet without GCG+ cells? At this point, I would guess that it is the minority! Indeed, this view is supported by the data in Fig 4G (and extended data Fig 7E – though I think INS+GLU- cells are mislabelled?) wherein only about 15 % of the total islet volume comprises GCG- islets. The degree to which insulin secretion is regulated by locally released glucagon (or indeed GLP1) is still a matter of debate. I think this finding needs to be toned down.

We understand the reviewers point but do not fully agree in that it would be better to ask how many beta cells belong to an islet without GCG+ cells (although both numbers and total volume contribution are given in the manuscript). Indeed, the degree to which insulin secretion is regulated by locally released glucagon or GLP1 (now added in the sentence) is not fully understood. However, little is known regarding the *in vivo* functionality of small vs. large islets in relation to GCG content. Existing data (although chiefly obtained *ex vivo* and/or in rats) suggests that smaller islets have a better insulin secreting capacity (see e.g., Lehmann et al Diabetes 2007 and McGregor et al., Am J Physiol Endocrinol Metab. 2006). We are also very clear about that it is primarily islets of the smaller size categories that are devoid of GCG+ cells (“Instead, the human endocrine pancreas comprises ~50% INS⁺ islets that belong to the smaller end of the spectrum of islet size ranges “see line 250-252). In our view, we currently don’t know enough about how islets of different hormone-cell build up (endocrine cell ratios) to judge the importance/impact of heterogeneities in islet size and cellularity for their capacity to respond to different metabolic stimuli (glucose, FFA, amino acids etc.). This is however an important task for future investigations and the full significance of the discovered existence of a large amount of GCG- islets (>50% of all INS⁺ islets) remains to be determined.

Figure S7E is indeed mislabeled and we thank the reviewer for pointing this out. This has now been corrected in the revised manuscript (New **Fig. S9**).

R2-2. Fig. 2 It be helpful to show summary data for all five pancreata analysed. We are left without a sense of variation in in islet size, cellularity etc. between donors.

In the manuscript, we show analyses for the percentage of INS⁺GCG⁻ present in all donors (see Table S1). In the revised manuscript we have further added new **Fig. S4**, which display; A, the INS⁺ volume/anatomy volume for one disc each from region 1-4 and, B/ INS⁺ volume/anatomy volume per size category in all 5 investigated pancreata. Further, we have shown individual values for INS/GCG ratios (new **Figs. S12 and 13**)

R2-3. I wasn’t always convinced that the authors provide the most straightforward explanation of their findings, e.g. lines 211-213. Surely a paucity of smaller islets after islet pancreatic disruption and isolation is likely to be due – at least in part - to their loss by digestion/mechanical damage?

We fully agree with the reviewer and the reference provided is wrong. We meant to compare our figures to those obtained of islets *in situ* and the reference has now been changed (see line 236 of the discussion).

R2-4. How do the authors explain the differences in islet distribution (more islets in the tail in the studies from Wang et al, 2013?) Could there be differences in the subjects used (age, MBI, ethnicity) or other technical issues (pancreas and data treatment)?

As the reviewer highlights, our data does not support the case where there is a higher islet density in the pancreatic tail based on insulin expression when analysing an entire pancreas, although variations exist between individual discs (Fig 2A). Further, analyses of individual discs from regions 1-4 in four additional pancreata (see new **Fig. S4** showing; A, the INS⁺ volume/anatomy volume for individual discs from region 1-4 and, B/ INS⁺ volume/anatomy volume per size category in all 5 investigated pancreata) do not lay bare an obvious trend towards increased BCM in the tail based on INS labelling. Of note, in other studies, by e.g., in Ionescu-Tirgoviste et al., Sci Reports 2015, islet area was estimated (based on H/E) staining's) on 5423 islets in total whereas in the study by Wang. et al., PLoS One, 2013, it was derived from 2D data obtained from 5µm paraffin sections in which the area and frequency of four hormonal cell types were investigated. In the latter study, it is unclear to us exactly how many islets were investigated and the sampling frequency. In our study, on the other hand, we have analysed the 3D volume of 2.21x10⁶ INS⁺ islets in one entire pancreas and in discs from region 1-4 from four additional donor pancreas, each disc encompassing around 30 000 INS⁺ islets. Indeed, all of the parameters, age, BMI, ethnicity etc., mentioned by the reviewer, together with differences in analytical methods (in particular regarding the amount of material analyzed together with the "absolute" versus "extrapolated" nature of the data and how it was normalized), are plausible to contribute to this difference. Further, as mentioned in the response to rev. #1, R1-18, the regions of the human pancreas are quite loosely defined into head, neck, body, tail (in particular between the latter two) in material from other studies. In the study by Wang. et al., PLoS One, 2013, islets were measured in the "tail at the end of the pancreas". Of note, a comparison of the donors analysed in the study by Wang et al., points to pronounced variations between head, body and tail between different donors (see figure 3 of that paper). We have in the revised manuscript made a note on this matter in the discussion (lines 232-234).

R2-5. I was missing the comparison to mouse pancreata in terms of the proportion of islets that are GCG-. Presumably published in an earlier report from this lab?

A main incentive behind the study was to provide an account of the β-cell mass across the entire human pancreas. By itself a significant undertaking. With this data at hand, we realized that a comparison with rodent β-cell mass distribution was justified. Such data has already been disseminated by our group in different studies (Parween et al., Sci Data 2017 and Hahn et al., Sci Data 2022, see accompanying Dryad links in the latter manuscript) and was therefore already available to us (and other researchers for statistical assessments/image analyses). As already outlined in the manuscript, we have in the present study used data from Hahn et al., Sci Data 2022 and have not performed any specific staining's of mouse tissue for this purpose. We have not previously generated comprehensive data on GCG combined INS distribution in the mouse. This in our view must be considered as a study by its own right and, as a note we are in the process of setting up experiments aimed at analyzing a wide variety of

markers in larger cohorts of mice, incorporating other endocrine cell-types together with markers for vascularization and innervation.

R2-6. I wonder why other islet cell types e.g. Sst+ were not measured?

As mentioned above, the main incentive behind the study was to provide an account of the β -cell mass across the entire human pancreas. This has allowed us to optimize methodology and protocols throughout the described pipeline. We are in the process of setting up an experiment in which all the major endocrine cell types will be imaged in pancreata from (to start with) non-diabetic donors. For this purpose, we will need to collect and incorporate other donor material to accommodate the combination of different epitopes. This however, is in our view, a study by its own right for which our current study will be a valuable resource to be used as a baseline, and of course also for more-clinically relevant analyses in the future.

R2-7. The study would be significantly reinforced by examination of pancreata from subjects with type 1 or type 2 diabetes.

We fully agree with the reviewer and this is certainly something we plan for the future. It will however require the collection of additional donor material on our hand. As a side note to the reviewer, we are in the process of analyzing, a so far limited, material from T1D donors, but again this must be considered as a study by its own right. Please also see answer directly above, R2-6.

Reviewer #3 (Remarks to the Author):

Professor Ahlgren has been pioneering optical projection tomography imaging of the pancreas, with studies providing valuable insights into changes occurring to beta cells - both in T1D and T2D. While most of his laboratory's studies were performed in rodents, a few years ago they provided the first complete OPT analysis of the human pancreas. They showed the 3D distribution of islets within human pancreata from non-diabetic and T2D donors. In the present manuscript they extend their studies by documenting the distribution of alpha cells and their co-presence with beta cells in individual islets throughout the entire human pancreas. One of the main findings is that a relatively large number of islets are devoid of alpha cells, which has important implications in terms of biological understanding of endocrine pancreas biology and further sheds light on islet heterogeneity. This is a significant finding that requires to be strongly supported by the author's data and analysis. In this respect my main questions are:

R3-1. - The identification and quantifications of ROIs are highly dependent on intensity thresholds used during image processing. This could artificially lead to an underestimation of the alpha cell population for instance, which would have important consequences on the conclusions of this study. How did the authors confirm that thresholds used for both insulin and glucagon signals are reflecting the real number and ratio of these cells within the human pancreas?

Since signal intensity can vary depending on where each ROI was located on the horizontal axes of each specimen (due to the configuration of the light sheets in the scanner), thresholds were carefully set manually (assessing both over- and under

exposed values) to reflect the borders of the labelled cells. Hence, the LSFM analyses of the ROIs are not batched processed. A note of this has now been introduced in the methods section under “Determining insulin and glucagon composition in OPT and LSFM”. Individual INS⁺GLU⁻ islets, identified in the 3D scans, were further analyzed, section by section, in a slide scanner (**Fig. S14** in the revised manuscript) confirming the absence/presence of the respective cell type.

R3-2.- Please clarify how many samples were used for all data analysis. Also, it is unclear from the abstract and introduction how many pancreata were assessed for this manuscript. This information is summarized in Table S1. We have in the revised manuscript clarified this point in the introduction (see lines 61-62).

In addition I would like the authors to address these comments:

R3-3.- It is stated that this manuscript provides the first complete representation of beta cells throughout the human pancreas (for example lines 20, 66, 204), however this has been done in one of their previously published studies.

We assume that the reviewer refers to Hahn et al., *Comms Biol*, 2021. In that study we did not provide a complete representation of beta cells throughout the human pancreas. Instead, that was a proof of principle study in which we analyzed a limited material from a single ND and T2D donor, using a slightly different protocol.

R3-4.- The statement in the abstract, “50% of the human insulin-expressing islets are virtually devoid of glucagon-producing alpha cells” is misleading. As we understand later in the text, 16% of BCM is comprised of islets devoid of alpha cells. This is a much lower number and the authors should discuss the relevance of this finding for the entire endocrine pancreas function.

We understand the reviewer's point, also raised by reviewer #2. Please see this response, R2-1.

R3-5.- The authors previously assessed the beta cell mass and distribution in a human T2D pancreas, and correctly mention in the introduction that “diabetes is a disease that involves all islet cell types”. Did the authors assess alpha cell distribution in a human T2D pancreas and/or could discuss how their current finding could apply to what happens in T2D?

We fully agree with the reviewer that this would be a very interesting undertaking. However, we feel that this is a study in its own right (which would benefit from a further understanding of INS⁺GLU⁻ islet functionality). Our current study will be a valuable resource to be used as a baseline for future more-clinically relevant analyses. Please also see response to R2-6 and R2-7.

R3-6.- From their previously published study, it was found that there are higher islet density areas in the organ periphery. Was this confirmed in the present study?

In the study referred to, we analyzed a limited material from older donors (66 and 69 years of age), and in that paper we made the following statement about these high islet density areas: “*Commonly, they were localized in regions in which the acinar*

tissue appeared disrupted by fibrosis and/or adipocytes, be it in pancreata from ND or diabetic donors.”, and we further speculated that “These observations suggest that peripancreatic inflammation may be a common feature of the human pancreas.” The current material consists of donors aged between 20 and 45 years of age. In this material we have not been able to discern the same grade of pathological changes nor the same increase in islet density of the organ periphery.

R3-7.- The authors should include a comparison of their results with previous published studies assessing beta cell mass in the human pancreas (using other methodologies, such as histology or flow cytometry). Also, findings regarding the ratio of alpha/beta cells should be compared to the general statement that beta cell content in human islets represents approximately 40-60% of all islet endocrine cells (see for example the review from Campbell et al., 2021, Nat Rev Mol Cell Biol).

We have in the manuscript given several references to studies calculating β -cell mass by different techniques, and to the ratio between alpha and beta cells and a report that has taken several studies into account (Ionescu-Tirgoviste et al., Sci Reports, 2015 and references therein). All of these are based, to a greater or lesser extent, on sampling techniques. The value given in Campbell et al., is 30-50% alpha cells and 50-60% beta cells but no reference is given in that paper as from where these figures are derived. Similar figures, $53.9 \pm 2.5\%$ β cells, $34.4 \pm 2.5\%$ α cells, are given in Brissova et al., Journal of Histochemistry & Cytochemistry 2005 but these are based on CLSM data of isolated islets. Indeed, it has previously been demonstrated that the human islet population is heterogenous with regards to cellular architecture and composition, (Miranda et al., Am J Physiol Endocrinol Metab320: E716–E731, 2021 and Kim et al., islets 2009) and different subtypes of endocrine cells even exist within individual islets (Dybala and Hara diabetes 2019, Miranda et al., Am J Physiol Endocrinol Metab320: E716–E731, 2021). In the revised manuscript, we have clarified this point and added relevant additional references (see lines 60-61), and now also mentioned in the discussion of the revised manuscript that: “It is of course possible that other parameters such as age, BMI, ethnicity etc., together with differences in analytical methods and the amount of material analyzed and how it has been normalized may account for differences between different reports on islet distribution and cellularity“ (lines 231-234).

R3-8.- In the Materials and Methods part, experiments on mouse pancreas are missing.

As stated in the manuscript, it is derived from another study (Hahn et al., Communications Biology 2020, Hahn. Et al., Scientific Data 2022) in which the utilized collection procedure is clearly referenced. In the revised manuscript, adequate references to the mouse pancreas isolation and processing have been added in the supplementary information under the subheading “Assessment of insulin islet 3D distribution and organization”. Please also see answer to reviewer #1, R1-18.

R3-9.- Figure 2: the average INS+ volume / anatomy volume is 3%, this should be compared to the literature. For consistency in the headings, the authors should decide between “volume”, “vol.”, “vol”. Panels A and B, Region 1: there are more data points presented in B than in A, could you explain what the individual data

points represent? Finally, in panel A, although this is explained in the text, the outlier bar should be marked differently (with “*” reserved for statistics). The bar itself could be distinguished as outlier by a different color and/or pattern.

Our value (2.8%, see figure legend Fig. 2) is now compared with other studies (219-20). If studies giving the amount of β -cells presented in weight (not a percentage) are excluded, most reports provide values of β -cell content in the pancreas in the range 1-3% (very often 1-2% without providing further references), see for example Rorsman and Ashcroft *Physiol Rev.* 2018 (and refs therein), Ichise and Harris, *Journal of Nuclear Medicine* July 2010, Demine et al., *Int. J. Mol. Sci.* 2020 and Weir and Bonner-Weir, *Ann. N.Y. Acad. Sci.* 2013. Hence, our value is within the upper range of this interval.

Very are grateful to the (very observant) reviewer for pointing out that there are more data points in B than there are discs in A. The discrepancy comes from that a few discs in the head region were cut in two (top-bottom) before scanning to fit these into the field of view of the scanner. These are now combined in new Fig. 2B (to constitute single data points) and a comment about it has been made in the methods section.

Further, in the revised manuscript the outlier in Fig 2A has been colored grey (explained in the figure legend) and the “*” has been removed.

R3-10.- Figure 3: was one human pancreas analysed in panel B, and five in panels H and I? Is the analysis on mouse pancreas based on 5 samples for D, for comparison with B? Since this panel compares distribution and sizes of islets in human versus mouse, the same information should be given for both species: panels E-I should be followed by similar panels for mouse. This would support the statement on lines 656-658.

We agree with the reviewer, and we have in the revised manuscript calculated panel B based on representative discs from regions 1-4 from all 5 donors (New **Fig. 3 B**). The mouse data in panel D is as before based on the entire pancreas from 5 mice. Further, we have added similar panels as in E-I, also for mice. See new **Fig. 3 J-N**.

R3-11.- Figure 4, panel F, what does each individual data point represent? From this panel, about 15% of islets contain alpha cells and are devoid of beta cells, is this correct?

Each data point represents the percentage of INS^+ and GCG^+ cells in the islets per Region of interest (ROI) (see methods). As stated in the figure legend: “Data are derived from ROIs from regions 1-4 from five different ND donor pancreata (see Table S1). Indeed, the data suggests that 16% of the islets are GCG^+INS^- . These, however only constitute a fraction (3%) of the total islet volumes and mostly consist of individual GCG^+ cells that cannot be delineated as part of an islet using the implemented method (see new **Fig. S12 and S13** and accompanying legend).

R3-12.- Figure 4, panel H, data should be clearly labeled as in panels F and G (and include $INS-GCG^+$) Also, when looking at the size category 50-100 in Fig 2F, there

are a total of approx. 8 insulin positive objects per mm³, contrasting with a total of about 20 objects in Fig 4H at the same size category, could you explain this discrepancy?

This is a good point and we believe there are two explanations to this. Firstly, the data in Fig. 2F are based on the INS⁺ islet population in an entire pancreas (H2457) as measured by OPT. The data in fig 4H, on the other hand, is based on LSFM data on ROIs from regions 1-4 from this pancreas and four additional donors. Hence, 8 INS⁺ islets/mm³ (in 2F) refers to the average from the total volume of 1 whole pancreas assessed by OPT, whereas the graph in Fig. 4H displays an average from in total 40 ROIs (115,6 mm³ of tissue from Region 1-4 from all five pancreata). This would, with the implemented step size of 5μm, in z, translate to about 8000 sections being 1.7x1.7mm in x-y. Still, it constitutes a fraction of the total volume investigated in the whole pancreas by OPT. Not only are there variations in INS/GCG ratios between the ROIs (see **Fig. S11-13**) as determined by LSFM, but there is also a variation of the number INS⁺ islets per size category between the pancreata and discs (see new Fig. S4). Further, the datasets are derived by two different technologies, which with the implemented settings for the current specimens generates different resolution (21μm for OPT and 1.9μm for LSFM). Whereas OPT generates isotropic voxels, LSFM displays elongation effects in the Z-axis due to the principal of detection by the scanner. Therefore, smaller GCG and INS volumes, otherwise under the threshold, could to an equal amount be included in the islet count in the LSFM analyses. Hence, this is likely to be attributed to a combination of variations between the analyzed pancreata/individual ROIs (for which the LSFM data covers a significantly smaller volume than the OPT data but instead provides an average from multiple donors) in combination with differences in the detection technology. In the revised manuscript, a note of this has been introduced into the discussion, see lines 308-313).

In the revised manuscript, panel 4H is now labelled as suggested by the reviewer. Further, we have for clarity exchanged Panel 4K with a graph showing the average INS/GCG ratio per size category (see also new **Fig. S13**).

R3-13.- Figure 4, panel J, the average islet content in alpha cells is 10-20% in the present study, which is low as compared to previous published studies (again, see for example Campbell et al., 2021, Nat Rev Mol Cell Biol). Can this be explained?

Please see response to R3-7, above.

R3-14.- Concerning the average alpha:beta cell ratio, a combined value is presented in Figure 4. Ratios for each individual donor should be provided in addition to this combined value. Ideally, ratios should also be provided individually per size category.

Ratios are now provided for individual specimen, please see new supplementary **Figs. S12 and S13**.

Finally, some minor points:

R3-15.- For consistency, the authors should use beta cell mass (BCM) and not BCV throughout the text (for example lines 85, 114, 126, 138).

This has now been corrected in the revised manuscript.

R3-16.- Line 38, “[...] islet volume of 0.5-2.0 cm³.”, please provide reference.

The reference was included a few lines down and in the revised manuscript it is given directly after this statement (Gabriela Da Silva Xavier *J Clin Med.* 2018 Mar; 7(3): 54. and references therein)

R3-17.- Line 82 and elsewhere, “and Fig. 3” is confusing as it appears to direct to Fig. 3 and not to the supplementary figure.

We are not exactly sure what the reviewer means here. The sentence reads: “By aligning the resultant tomographic datasets together in 3D space, an entire pancreas could be “rebuilt” with regards to the 3D distribution of INS⁺ cells (**Fig. 1D, Movie S2 and Fig. S3**)”. Fig. S3 is a higher resolution image of the “rebuilt” pancreas which was the intention.

R3-18.- Line 118, could you confirm the this is “average” and not “median”?

This is average

R3-19.- Line 119: the correction for tissue processing is confusing here. Is it corrected elsewhere or are all other values, including in the figures, left uncorrected? How the corrected value was calculated should also be explained or a reference provided.

All the values presented in the report are uncorrected and the estimated value (75μm) was estimated based on typical tissue shrinkage using the implemented tissue processing protocols (in the range of 10-15%, see e.g., Winsor L. Tissue processing. In Woods and Ellis eds. *Laboratory histopathology.* New York: Churchill Livingstone, 1994;4.2-1 – 4.2-39, where we used the upper range, 15%). However, we are grateful to the reviewer for emphasizing this matter since tissue shrinkage is tissue dependent. We have therefore in the revised manuscript included a new analysis illustrating the degree of tissue shrinkage in the current material (see **new Fig. S7**) and clarified this in the text (see lines 122-126). In fact, this analysis shows a smaller shrinkage (~5%), resulting in an even lower compensated value for the average islet diameter of 68.3μm. This value is now introduced in the revised manuscript. Of note, tissue processing for e.g., paraffin sections carries similar effects.

- **R3-20.** Line 146, “(see above)”, which other reports are meant? Should it be a reference?

Adequate references are now in place (see line 154).

R3-21.- Lines 151-153, “[...] and in contrast [...]”, it is not clear what the authors mean.

The sentence lacks a word and should read “in contrast to previous stereological assessments we could not find convincing evidence for islet routes....”. or varying islets densities in specific areas of the pancreas. This has now been corrected.

R3-22.- Lines 159-161, “the average islet size is significantly smaller than what has been reported by 2D stereological studies”, please provide data and references.

Data and adequate references are now inserted directly after the statement (see lines 166-171).

R3-23.- Lines 161-162, “human beta cell mass organization differs significantly from that of the mouse”, please provide data to support this conclusion.

This whole section is a summary statement of the data provided above where adequate references are now in place.

R3-24.- Line 175, “consensus 2:1 beta cell to alpha cell ratio”, and line 220, please provide reference(s).

References are now provided in direct conjunction with the statements.

R3-25.- Line 249, “small islets have superior cellular function compared with larger islets”, the reference 30 regards islet transplantation and not islets in situ.

We thank the reviewer for pointing this out. The wrong reference was erroneously inserted. We have now exchanged the reference and the sentence now reads: “*in vitro* studies, of rats and human islets, suggest that smaller islets have superior cellular function compared with larger islets” (Lehmann et al Diabetes 2007 and McGregor et al., Am J Physiol Endocrinol Metab. 2006, Farhat et al., Islets 2013).

R3-26.- Line 251, small mouse islets have been reported to be almost devoid of alpha cells. This might be due to the isolation procedure and not a confirmation of the present findings (furthermore, mouse islets have a much lower percentage of alpha cells).

The sentence reads ..”been demonstrated that small mouse islets after isolation are virtually devoid of α -cells.” We agree with the reviewer, and we do not claim this. It is a possibility mentioned in the discussion section that would need additional experiments in mice to substantiate.

R3-27.- Figure 1D, and Extended Data Fig. 3 are identical and almost of same size, why is there an Extended Data Fig. 3?

They are indeed the same, but whereas Fig. 1D is included for illustration of the data generation process, Fig. S3 is included as a higher-resolution image of the complete BCM distribution of a human pancreas without labelling (boxes, arrows, lettering etc.) that would otherwise disturb the picture. We are happy to remove it on the editor’s discretion.

R3-28.- Extended Data Fig. 6, panel D: large islets seem to be more present in the periphery of the sample, is this due to the sample processing/staining procedure? Legend for panels Q-R, images are representative of how many mouse pancreas samples?

This phenomenon in our view could be attributed to a number of factors and is something we see in a few discs. In part it could be caused by an optical effect caused by the angle of view. I.e., how the sample is tilted. The inserted images show the same sample as in Fig. S8D (old Fig. S6 panel D) in different angles. It may also in part be related to illumination intensity distributions over the field of view (to find a balance between sample size and the practical possibility to cover the entire volume of a human pancreas, the “discs”, even for OPT constitute samples of significant dimensions). Finally, it could be a part of natural variation within the pancreas, see e.g., panel P, which is scanned and post-processed exactly the same way as the disc in D.

The images are representative for 5 mouse pancreata (now clarified in the figure legend).

R3-29.- Extended Data Fig. 11, all samples should be presented individually, rather than having combined values for H2456, 2457, 2466, 2522.

New images showing values for the individual donors are now introduced as **Figs. S12 and S13**.

R3-30.- Extended Data Fig. 12 is not referred to in the main text.

Fig. s12, now **Fig. S16**, is rereferred to in the Methods section of the Supplementary information.

Despite all of the above, this is an impressive study providing essential information

on the composition of the human pancreas at high resolution. I am looking forward to the revised manuscript.

REVIEWERS' COMMENTS

Reviewer #1 (Remarks to the Author):

I appreciate the detailed and rerally comprehensive set of replies to the variety of concerns I pointed out during the first review. I appreciate the efforts of the authors regarding the new supplementary figures, and I believe the new details included in the new version of the paper will be extremely helpful in improving the paper's readability and the general public's understanding of the paper itself. I personally appreciate the detailed explanation of points which were not very clear to me, and I am grateful to the authors for their thorough and exhaustive explanation. The new references added are important to acknowledge previous works and allow better understanding of certain importanto points discussed in the paper. The different notes and new extended figures about resolution and fixation protocols are also crucial for the readability of the paper by non-experts in the field, such as the authors. I think the paper has been improved very much for the previous version and I appreciate all the efforts of the authors. Although I did not asked for these experiments in this paper, I strongly look forward to the authors future efforts in discussing and understanding the significance of the islet's compositional heterogeneity on islet function through gene expression or sequencing or any other technique combined to the imaging part.

Reviewer #2 (Remarks to the Author):

I would like to congratulate thje authors on a very thorough response to my own (and the other reviewers`) queries.

Nevertheless, I was slightly puzzled that they have insisted on leaving lines 250-253 unchanged. I still feel that the bald statement that "the human endocrine pancreas comprises ~50% INS+ islets...." is likely to be mis-interpreted. The sentence goes on (rightly) to say that these ".. belong to the smaller end of the 252 spectrum of islet size ranges ... "

I would urge the authors to avoid stirring up controversy unncessarily - a risk which they can

reduce significantly by splitting this sentence, and starting the second by saying something along the lines of: "We would emphasise that these belong to the smaller end of the spectrum and, consequently, that most beta cells reside in an islet that possesses alpha cells.." or similar.

Once again, congratulations on a terrific piece of work.

Reviewer #3 (Remarks to the Author):

I would like to thank the authors for significantly improving their manuscript and for adequately addressing all my comments and concerns. This impressive work and the new findings will be of importance for the research community in the field of pancreas biology.

Response to referees

Lehrstrand et al.,

We would like to thank the reviewers for their constructive comments, which we feel has contributed to further strengthen our manuscript. We have addressed a single note regarding the re-revised manuscript in **green** (R2.1).

R2.1 Added in re-revision.

Reviewer #2 (Remarks to the Author):

I would like to congratulate the authors on a very thorough response to my own (and the other reviewers') queries.

Nevertheless, I was slightly puzzled that they have insisted on leaving lines 250-253 unchanged. I still feel that the bald statement that "the human endocrine pancreas comprises ~50% INS⁺ islets..." is likely to be mis-interpreted. The sentence goes on (rightly) to say that these "... belong to the smaller end of the 252 spectrum of islet size ranges ... "

I would urge the authors to avoid stirring up controversy unnecessarily - a risk which they can reduce significantly by splitting this sentence, and starting the second by saying something along the lines of: "We would emphasise that these belong to the smaller end of the spectrum and, consequently, that most beta cells reside in an islet that possesses alpha cells.." or similar.

Once again, congratulations on a terrific piece of work.

We feel that since the relationship between islet cellularity and islet function is not established in this case, and that we are already fully clear with that the GCG⁻ islets are belonging to the smaller size categories (as also acknowledged by the reviewer) this is a justified statement, stirring up controversy or not. Unless his/her note is mainly of semantic nature (in which he/she indeed may have a point), we feel that the suggested sentence could instead be misinterpreted the other way. I.e., that this large pool of INS⁺GCG⁻ islets would be less significant or important, which we do not know at present. However, to make this clearer we have in the re-revised manuscript changed the sentence on lines 250-253 (now 249-252)

"Instead, the human endocrine pancreas comprises ~50% INS⁺ islets that belong to the smaller end of the spectrum of islet size ranges (still larger than 29 μm in diameter), which are essentially devoid of GCG⁺ cells."

So that it now reads:

"Instead, the human endocrine pancreas comprises ~50% INS⁺ islets, which are essentially devoid of GCG⁺ cells. These are however predominantly found in the smaller end of the spectrum of islet size ranges (still larger than 29 μm in diameter)."